# Does the East Greenland Current exist in northern Fram Strait?

Maren Elisabeth Richter[1], Wilken-Jon von Appen[1], and Claudia Wekerle[1]

[1]Alfred Wegener Institute, Helmholtz Centre for Polar and Marine Research, Am Handelshafen 12, 27570 Bremerhaven, Germany

**Correspondence:** Maren Elisabeth Richter (maren.richter@awi.de)

**Abstract.** Warm Atlantic Water (AW) flows around the Nordic Seas in a cyclonic boundary current loop. Some AW enters the Arctic Ocean where it is transformed to Arctic Atlantic Water (AAW) before exiting through Fram Strait. There the AAW is joined by recirculating AW. Here we present the first summer synoptic study targeted at resolving this confluence in Fram Strait which forms the East Greenland Current (EGC). Absolute geostrophic velocities and hydrography from observations in 2016, including four sections crossing the east Greenland shelfbreak, are compared to output from an eddy-resolving configuration of the sea-ice ocean model FESOM. Far offshore (120 km at 80.8° N) AW warmer than 2° C is found in northern Fram Strait. The Arctic Ocean outflow there is broad and barotropic, but gets narrower and more baroclinic toward the south as recirculating AW increases the cross-shelfbreak density gradient. This barotropic to baroclinic transition appears to form the well-known EGC boundary current flowing along the shelfbreak farther south where it has been previously described. In this realization, between 80.2° N and 76.5° N, the southward transport along the east Greenland shelfbreak increases from roughly 1 Sv to about 4 Sv and the proportion of AW to AAW also increases fourfold from 19±8% to 80±3%. Consequently, in southern Fram Strait, AW can propagate into Norske Trough on the east Greenland shelf and reach the large marine terminating glaciers there. High instantaneous variability observed in both the synoptic data and the model output is attributed to eddies, the representation of which is crucial as they mediate the westward transport of AW in the recirculation and thus structure the confluence forming the EGC.

## 1 Introduction

Fram Strait, located between Greenland and Svalbard, is the only deep connection between the Arctic Ocean and the Nordic Seas. Pathways and modification of watermasses there and on the northeast Greenland shelf are insufficiently understood. The northeast Greenland shelf is dominated by a C-shaped trough: Westwind Trough and Norske Trough cross the shelfbreak at ~80.5° N and at ~76.5° N respectively (Fig. 1). These allow exchange flows below 100 m depth between the outlet glaciers of the North East Greenland Ice Stream (NEGIS), the largest of which is the 79N Glacier, and the deep Fram Strait. Thus, the troughs may provide a pathway for warm, saline waters of Atlantic origin to the glaciers where they contribute to submarine melt (Schaffer et al., 2017).

Relatively warm (warmer than 2°C) and salty Atlantic Water (AW) enters the Nordic Seas across the Greenland-Scotland Ridge and flows in a cyclonic boundary current loop (Mauritzen, 1996) comprised of the northward flowing Norwegian Atlantic Current and West Spitsbergen Current (WSC) in eastern Fram Strait (see Hanzlick, 1983, for a review of early studies of

the WSC), and the southward flowing East Greenland Current (EGC, see Aagaard and Coachman, 1968, for a review of early observations of the EGC) in western Fram Strait. During the loop through the Nordic Seas AW is cooled and densified, forming part of the Denmark Strait Overflow Water (DSOW) which becomes the dense component of the North Atlantic Deep Water south of the Greenland-Scotland Ridge (Mauritzen, 1996; Rudels et al., 2002; Våge et al., 2013). Some AW enters the Arctic

Ocean through the Barents Sea and Fram Strait. AW flows cyclonically around the Arctic Ocean (Aksenov et al., 2011), where it is transformed to Arctic Atlantic Water (AAW), which is fresher and colder than AW (Schauer et al., 1997; Rudels et al., 2005, 2012), before exiting the Arctic Ocean via Fram Strait. The Arctic Ocean halocline separates the warm salty intermediate waters of Atlantic origin from colder and fresher Polar Surface Water (PSW) at the surface (Rudels et al., 1996, 2005). Some of the AW is transformed into halocline water (e.g. Rudels et al., 2015).

The WSC is a complex current with barotropic and baroclinic components, that splits into multiple branches (Quadfasel et al., 1987) and produces eddies (von Appen et al., 2016). At 79° N the WSC is made up of two branches, the shelfbreak branch and the offshore branch. The latter is stronger in winter than in summer and mostly baroclinic (Beszczynska-Möller et al., 2012). The zonal section across Fram Strait most frequently occupied lies at 78° 50'–79° N. An array of moorings along this line has made long term observations possible. Both the mooring array and summer synoptic surveys of the WSC show a warming of

the watercolumn since the mid 1990s (Beszczynska-Möller et al., 2012; von Appen et al., 2015; Walczowski et al., 2017). Half to two thirds of the AW flowing northward in the WSC at 79° N recirculates from the eastern boundary currents in Fram Strait to the EGC in the west (Rudels, 1987; Manley, 1995; Marnela et al., 2013). North of 79° N the WSC splits into three branches: the Svalbard and Yermak branches (Perkin and Lewis, 1984) and a recirculating flow of AW (Gascard et al., 1995). The Svalbard branch was supposed to be the main flow of AW into the Arctic (Manley, 1995) though a pathway crossing the Yermak

Plateau at the Yermak Pass (Gascard et al., 1995) was recently proposed as the main AW inflow to the Arctic Ocean (Koenig et al., 2017). The Yermak Branch partly recirculates in northern Fram Strait though there is no agreement in the literature on the exact amount and location (e.g. Aagaard et al., 1987; Manley, 1995; Schlichtholz and Houssais, 1999; Hattermann et al., 2016).

The westward transport of AW from the WSC to the EGC was first proposed by Ryder (1895, pg. 204) and has variously

been described as a topographically steered branch of the Greenland Sea Gyre in the southern part of Fram Strait (following the Knipovitch Ridge and the Greenland-Spitsbergen Sill), as a cyclonic circulation around the Molloy Hole seen in models (Aksenov et al., 2010; Kawasaki and Hasumi, 2016; Hattermann et al., 2016; Wekerle et al., 2017) and observations (Johannessen et al., 1987; Quadfasel et al., 1987) and as a field of topographically steered eddies, shed by the WSC, which merge with the EGC (Gascard et al., 1988, 1995). These eddies are also thought to be one mechanism that allows the AW to subduct

underneath the sea-ice and PSW advected from the Arctic Ocean southwards in the EGC (Hattermann et al., 2016). The wintertime peak in eddy kinetic energy (EKE) observed in Fram Strait (von Appen et al., 2016) can be explained by the greater baroclinic and barotropic instability of the WSC in winter compared to summer (Teigen et al., 2010, 2011) due to decreased stratification of the upper watercolumn (von Appen et al., 2016). An eddy resolving grid is required in numerical models to reproduce the observed EKE levels (Hattermann et al., 2016; Wekerle et al., 2017). This increases AW transport into central

Fram Strait and vertical transport of AW i.e. subduction under sea-ice and PSW. As described in Wekerle et al. (2017), a correct

eddy heat transport is also able to reduce the cold bias (a common bias in ice-ocean models (Ilicak et al., 2016) where modelled temperatures are lower than in observations) in FESOM found in the central Fram Strait.

The meridional extent of the recirculation is at present unclear. The location of the northern rim of the recirculation probably depends on the watermass tracked as well as the time of the measurements. Some observational studies locate the northern rim south of 81° N (Rudels et al., 2005). There is evidence from drifter data (Gascard et al., 1995), hydrographic surveys (Marnela et al., 2013), an inverse modelling study (Schlichtholz and Houssais, 1999) and a numerical ocean model (Kawasaki and Hasumi, 2016) that the recirculation in Fram Strait may extend beyond 81° N, possibly as far north as 82° N. However, evidence from model studies in Fram Strait is at present inconclusive as the northern limit of the recirculation, the strength of individual recirculation pathways and of the boundary currents varies between models (Maslowski et al., 2004; Aksenov et al., 2010; Hattermann et al., 2016; Ilicak et al., 2016; Wekerle et al., 2017). This appears to be related to the resolution of the models and the bathymetry (Fieg et al., 2010). Observations able to determine the strength and location of the recirculation are therefore needed.

However, due to heavy sea-ice conditions observational studies in the central and western Fram Strait significantly north of 79° N are scarce. Thus, the northern AW recirculation and EGC remain undersampled and poorly understood. A study of the Arctic Ocean outflow along the northeast Greenland shelfbreak using data from 82–83° N (Falck et al., 2005) shows no recirculating AW there. South of 79° N, the EGC is a current located offshore of the Greenland shelfbreak on the western side of Fram Strait that transports recirculated AW and modified AAW below relatively fresh and cold PSW and sea-ice from the Arctic (Aagaard and Coachman, 1968). Both recirculating AW and AAW lose contact with the atmosphere before reaching northern Fram Strait. The different transit times through the Arctic Ocean (~1 year to 10s of years: Karcher et al., 2003; Polyakov et al., 2011) compared to the recirculation in Fram Strait (on the order of months for AW: Gascard et al., 1995; Hattermann et al., 2016) have also been inferred from the lower oxygen saturation of AAW compared to AW. Between 78° N and Denmark Strait the EGC consists of three branches: an inshore branch transporting fresh cold water, a shelfbreak branch and a branch offshore of the shelfbreak believed to be a direct recirculation of AW from the western WSC branch (Håvik et al., 2017; Nilsson et al., 2008; Woodgate et al., 1999).

The aim of this study is to utilize the first synoptic data set targeted at investigating the structure of the EGC and of the AW recirculation in Fram Strait north of 79° N. We will describe the hydrography (potential temperature, salinity, potential density) and the kinematics (absolute geostrophic velocity fields) along the path of Atlantic Water (AW) in Fram Strait. We start with the inflow of AW and the WSC at 79° N in Sect. 3.1. Then, we turn to central Fram Strait and the westward recirculation of AW crossing the prime meridian (0° EW; Sect. 3.2) before we follow the path of the southward flow along the east Greenland shelf from ~80.3° N to 76.6° N in Sect. 3.3. We will examine the formation and transport of the EGC in Sect. 4.1 and take a look at shelf processes on the northeast Greenland shelf in Sect. 4.2. Throughout, we utilize an eddy resolving numerical model (Wekerle et al., 2017) to put the synoptic observations in a larger temporal and spatial context and to assess in which state of the highly variable flow regime the observations were taken. We close with conclusions from our findings in Sect. 5.

## 2 Data and methods

### 2.1 CTD and ADCP data

Data was collected between 18[th] of July and 6[th] of September 2016 during cruise PS100 of RV *Polarstern*. The data consists of 75 stations along 5 sections (0° EW, 79° N, WT1, 79.6° N and NT1; Fig. 1). CTD casts (Kanzow et al., 2017a, b) were

recorded with a dual duct Seabird 911+ and averaged into 1 m bins (Kanzow, 2017). The conductivity and oxygen sensors were calibrated using water samples analysed on-board (Kanzow, 2017) with an Optimare Precision Salinometer and with a titration method, respectively. An upward and a downward looking 300 kHz RDI Workhorse Acoustic Doppler Current Profiler (ADCP) were used as a lowered-ADCP (LADCP) system (von Appen et al., 2017). A vessel-mounted 150 kHz RDI Ocean Surveyor ADCP (VMADCP) recorded ocean velocities along the cruise track (Kanzow and Witte, 2016). ADCP velocities

were detided by subtracting the barotropic tidal component calculated from the Arctic Ocean Tidal Inverse Model (AOTIM-5, Padman and Erofeeva, 2004). VMADCP and LADCP setup and processing are described in detail in Kanzow (2017).

### 2.2 Data processing

For each section, station locations are projected onto the straight lines shown in Fig. 1 retaining their longitude (latitude in the case of 0° EW). Bathymetry information from the ship's echosounder, the IBCAO V3 bathymetry (Jakobsson et al., 2012) and

CTD altimeter station depths agreed to within 10s of meters. Therefore, we use the linearly interpolated station depths to plot the bathymetry in the sections. In section WT1 the location of the shelfbreak is corrected using the echosounder bathymetry. The easternmost bathymetry at 79° N near the Svalbard shelf is corrected using IBCAO bathymetry of the Svalbard shelfbreak. In section 0° EW we use the bathymetry from IBCAO for the entire section and interpolated hydrographical values appearing below the so-defined seafloor are removed before plotting.

For each CTD station the VMADCP velocity profiles are averaged whilst the ship was on station to attain a single profile. For each section the station data (CTD, LADCP and VMADCP) are interpolated onto a common grid with vertical resolution of 10 m and a horizontal resolution of half the mean station distance of the section (ranging from 5 to 20 km) using a Laplacian-Spline interpolation (Smith and Wessel, 1990). A standard tension of 5 (0 = Laplacian interpolation, $\infty$ = spline interpolation) and a search radius of 10 grid points are used. Geostrophic shear is calculated from the gridded hydrography using thermal

wind and is referenced to the 50-150 m averaged on-station VMADCP velocities (except for section NT1 where the 50–150 m LADCP data is used) to obtain absolute geostrophic velocities. For conceptual considerations, we additionally use a simple two layer ocean approximation with a density difference of $0.3 \text{ kg m}^{-3}$ to estimate baroclinic velocities from the slope of the $27.8 \text{ kg m}^{-3}$ isopycnal. The position, width and core velocity of the shelfbreak EGC and WSC are defined following Håvik et al. (2017): The core velocity is the maximum of the 0–150 m mean velocity of the section. The boundaries of the EGC and

WSC are defined as the locations where the 0–150 m mean velocity has decreased to 20 % of the core value. This criterion is also used to define the boundaries of the EGC within which we calculate net transport. It has the advantage over using a fixed width or distance from the shelfbreak that it can account for a meandering current of variable width, as we expect to see in synoptic observations.

To assess the errors due to the gridding process, the CTD and ADCP data are regridded increasing or decreasing a) tension, b) search radius and c) grid resolution individually by a factor of two. The relative absolute error of the absolute geostrophic velocity between the modified grid and the grid used in this study is determined. Velocity error estimates from a and b are generally below 10 %, with some higher values occurring below 500 m outside of the EGC at 79.6° N. Velocity errors from c are mostly below 30 %, higher values are found in areas of large and uneven station spacing. Note that a change in grid spacing of factor two is rather large and thus presents a maximum error estimate. The error of the VMADCP measurements is calculated as the median absolute deviation over the full sampling depth in time and space whilst on station and is ~0.04 $\mathrm{m\,s^{-1}}$ with maximum values of 0.07 $\mathrm{m\,s^{-1}}$. The processing routine for LADCP velocities gives an error estimate dependant on depth for each cast (Thurnherr, 2010; Kanzow, 2017). The median error between 50 and 150 m depth at section NT1 is below 0.05 $\mathrm{m\,s^{-1}}$ for all except for the easternmost station where it is 0.1 $\mathrm{m\,s^{-1}}$. Transport error estimates combine errors from calculating the reference velocity from the ADCP measurements, errors introduced by the tidal model during detiding, errors in calculating the geostrophic velocity from the hydrography and the effect of station spacing and the ship's drift on station. Errors from the tidal model are mainly due to inaccuracies in the bathymetry used in the model. We try to minimise these errors by taking the tidal transport calculated by the model and then calculating the tidal velocity with a more exact bathymetry. The combined transport error was calculated following Sutherland (2008).

## 2.3  Numerical model

In this study we use model output from the Finite-Element Sea-ice Ocean Model (FESOM) version 1.4 (Wang et al., 2014; Danilov et al., 2015). FESOM is an ocean-sea ice model which solves the hydrostatic primitive equations in the Boussinesq approximation. The sea ice component applies the elastic-viscous-plastic rheology (Hunke and Dukowicz, 2001) and thermo-dynamics following Parkinson and Washington (1979). The finite element method is used to discretise the governing equations, applying unstructured triangular meshes in the horizontal and z-levels in the vertical.

We use a global FESOM configuration that was optimised for Fram Strait, applying a mesh resolution of 1 km in the area 75° N–82.5° N/20° W–20° E and 4.5 km in the Nordic Seas and Arctic Ocean (Wekerle et al., 2017). In comparison to the local Rossby radius of deformation (around 4–6 km in Fram Strait, von Appen et al., 2016), this configuration can be considered as "eddy-resolving". The model bathymetry was taken from RTopo2 (Schaffer et al., 2016). The model is forced with the atmospheric reanalysis data COREv.2 (Large and Yeager, 2009), and interannual monthly river runoff is taken from Dai et al. (2009). The simulation covers the time period 2000–2009, and daily model output was saved. Model runs do not go beyond 2009 since the forcing dataset does not include more recent years.

Comparison with various observational data showed that the model generally performs very well in terms of circulation structure, eddy activity and hydrography (Wekerle et al., 2017) which makes us confident that we can use it as a best-estimate realistic hindcast of the circulation and hydrography in Fram Strait. However, there is a bias toward higher salinity in the Atlantic Water layer of around 0.15. This salinity bias can be traced back into the North Atlantic, and is a result of model deficiencies in correctly representing the pathways of the North Atlantic Current.

Eddy kinetic energy (EKE) is computed by decomposing velocities $u$ and $v$ into monthly means (denoted by bar) and a devi-

ating part (denoted by prime). The time-averaged EKE is then

$$\overline{EKE} = \frac{1}{2}\overline{((u')^2 + (v')^2)} = \frac{1}{2}(u^2 - \overline{u}^2 + v^2 - \overline{v}^2). \tag{1}$$

## 2.4 Watermass definitions and calculations

Watermass definitions (see Table 1) follow Rudels et al. (2005) except for very warm AW. Following Walczowski et al. (2017),

we include water lighter than $27.7\ \mathrm{kg\,m^{-3}}$ with salinities above 34.92 in our definition of AW. This definition ensures that surface water in the WSC is defined as AW. Additionally we define Denmark Strait Overflow Water (DSOW) as water above $800\ \mathrm{m}$ depth which is denser than $27.8\ \mathrm{kg\,m^{-3}}$.

The deep $\theta$ maximum is defined as the subsurface maximum in potential temperature, if this criterion is not sufficient it is defined as the depth of maximum salinity (Richter, 2017). Endmembers for mixing calculations are picked as the deepest water

sampled (DW), the warmest subsurface $\theta$ peak found in the AW inflow region at 79° N (AW), the coldest clearly defined deep temperature maximum (AAW) and the coldest water sampled (PSW) and are given in Table 2. Since AW and/or AAW is always located between PSW and DW, and since DW and PSW are not observed to mix, we can describe our observations as either AW-AAW-PSW mixtures or as AW-AAW-DW mixtures. The resulting mixing triangles are shown in Fig. 2. Note that the relative contribution of AW and AAW in a water parcel that is mostly comprised of AW and AAW is not affected by

this method. Errorbars for the watermass fractions are calculated by repeating the calculation 1000 times including random normally distributed uncertainties for the temperature and salinity of the endmembers with a standard deviation of 0.2°C and 0.04 PSU respectively. Please note that the distribution of uncertainties naturally includes values outside of the $\pm 1$ standard deviation boundary. The reported uncertainties correspond to the standard deviation over all realizations of the watermass calculation.

# 3  Results

We now present our results following the path of Atlantic Water through Fram Strait, from the inflow in the WSC, via the recirculation in central Fram Strait to the EGC. A particular emphasis is placed on the formation and evolution of the EGC.

## 3.1  The Atlantic Water inflow in the WSC

The most striking feature of section 79° N, as measured in summer 2016, is the highly dynamic velocity field (Fig. 3c). This

can also be seen in daily averages from FESOM (Supplementary Material: Movie S1b) and in the multi-year model average eddy kinetic energy at 79° N (Fig. 4c) which is significant across Fram Strait east of 5° W and highest over the Svalbard shelf slope. This agrees with observations (von Appen et al., 2016). The velocity field may be comprised of eddies, which appear as strong velocity fluctuations paired around domes in the temperature and density fields (Fig. 3). While the precise horizontal structure of these cannot be resolved here, the it matches that of the daily averages of the modelled velocity field

(Supplementary Material: Movie S1). It is clear that the flow is not smooth, i.e. unidirectional, in the WSC and EGC with near 0 velocities otherwise as seen in long-term mean sections (e.g. Beszczynska-Möller et al., 2012). Separate from the eddies, we

identify the northward velocities east of the 1000 m isobath on the Svalbard slope (Fig. 3c) as the WSC. This location agrees with the location of the WSC core both in long-term observations (Beszczynska-Möller et al., 2012) and FESOM output (Fig. 4b). The velocities in the eddies are instantaneously stronger than the WSC with peak velocities of -0.18 and 0.24 $\mathrm{m\,s^{-1}}$ (e.g. at 240 km and 260 km in Fig. 3c).

Whilst the 27.8 $\mathrm{kg\,m^{-3}}$ isopycnal (Fig. 3b) is almost flat in the deep Fram Strait (west of 2.5° E), near the Svalbard slope, it slopes downward toward the east with 0.64 $\mathrm{m\,km^{-1}}$. The downward sloping of isopycnals in the vicinity of the shelfbreak is a characteristic of baroclinic boundary currents, such as the WSC and EGC. The isopycnal slope is used to estimate the baroclinc velocity assuming a two-layer ocean as described in Sect. 2.2. This conceptual estimate gives a baroclinic velocity of 0.13 $\mathrm{m\,s^{-1}}$ in the WSC. Though only a rough estimate, this value is close to the absolute geostrophic velocity in the WSC of 0.11 $\mathrm{m\,s^{-1}}$ (Fig. 3c). We did not observe an offshore branch of the WSC which is consistent with long term measurements where the offshore branch is observed to be weakest or absent during summer months (von Appen et al., 2016; Beszczynska-Möller et al., 2012). Additionally, the presence of an offshore branch may be obscured by an eddy in our transect.

The water column in the WSC is temperature stratified with a temperature maximum at the surface, while the minimum temperature is in the deep ocean (Fig. 3a+b). The surface temperatures of over 9°C on the west Spitsbergen slope are the highest water temperatures in the WSC near 79° N published so far and are likely due to the warming of the AW inflow to Fram Strait (Beszczynska-Möller et al., 2012; Walczowski et al., 2017). The AW layer is over 500 m thick and is in contact with the atmosphere east of 5° E (Fig. 3a). Toward the west the AW layer gets thinner and the depth of the temperature maximum increases. Although water warmer than 2°C is found in the upper 50 m west of 5° E, this water is too fresh to fall into the AW definition (Fig. 3a/b).

## 3.2 The westward recirculation in the deep Fram Strait

The synoptic section in central Fram Strait shows a south to north transition along 0° EW. At the southernmost station (near 78° N) the water has an almost uniform salinity with warm AW close to the surface (the water in the upper tens of meters is too fresh to fall into the AW definition) and colder water at depth (Fig. 5a), similar to the stations sampled in the WSC along 79° N (Fig. 3a). With increasing latitude the observed AW layer gets thinner, colder, fresher and is located deeper in the water column. This suggests that between the AW inflow at the surface in the WSC and the subsurface AW layer in the northern part of the central Fram Strait, AW subducts underneath colder and fresher PSW and sea-ice. This was also simulated in the eddy resolving model study of Fram Strait by Hattermann et al. (2016) and it was hypothesized that baroclinic instability may achieve this subduction. The subduction of AW under PSW is also simulated in FESOM though this does not show a northward thinning of the AW layer (Fig. 5b). In the observations the Arctic Ocean halocline, with cold, fresh PSW at the surface, is found in the upper 120 m of the water column north of 80° N below which Knee Water (KW, the saltiest water close to the freezing point line) is found. The properties of KW are indicative of the ice-ocean-atmosphere interaction in the Arctic Ocean (Moore and Wallace, 1988; Rudels et al., 2005) signalling that we observe water modified in the the Arctic Ocean north of 80° N. In addition to their maximum temperature (more or less than 2°C) AW and AAW along 0° EW exhibit differences in oxygen saturation. Since AAW has transited through the Arctic Ocean, its oxygen saturation of typically ~80 % is significantly lower

than the oxygen saturation of AW of typically ~100 %.

AW is present somewhere in the watercolumn at all stations along 0° EW except for the northernmost station at 80.8° N (Fig. 5a). This implies that we sampled either the northern rim of the recirculation as it was at the time of our measurements, or that we sampled a passing AAW filament. We cannot decide which of the two explanations is true since no measurements farther north than 80.8° N were taken during the cruise. Examining the mean temperature in FESOM at 0° EW (Fig. 5b) shows average temperatures above 2°C at 80.8° N in central Fram Strait. This suggests that the northern rim of the recirculation in the model lies northward of this. Alternatively, the presence of warm water at this latitude in the model may be related to the presence of the Yermak branch flowing into the Arctic Ocean close to 0° EW. However, this does not agree with the modelled average velocities in the AW layer (Fig. 5d and 8c+d) which are southeastward north of ~80° N. AAW eddies with a temperature maximum below 2°C are seen in the daily averages of the model run for 2009 (Supplementary Material: Movie S2). Hence the model does not allow us to judge which of the two possible explanations is more likely. The synoptic observations made here do however show that the recirculation in Fram Strait can reach as far north as 80.7° N. A repeat synoptic survey along 0° EW, with a higher resolution than in the present study, extending beyond 81° N, supported by a mooring array, could provide a more definite picture of the northern limit of the Fram Strait recirculation and its meridional and temporal structure. This could then be used to verify numerical models.

In the central Fram Strait along 0° EW (Fig. 5c) the coarse resolution section depicts an absolute geostrophic velocity field which switches between broad sectors of weak eastward (~78° N and ~79.5° N) and westward velocity (~78.5 to ~79° N and around 80 to 80.5° N). Velocities reach $\pm 0.12 \, \mathrm{m\,s^{-1}}$. The velocity field appears mostly barotropic but the station spacing of ~40 km is not able to resolve the flow structure. We expect the velocity field, at least in the vicinity of 79° N, to be similar to the velocity field shown in Fig. 3c at 79° N and 0° EW. This is supported by the modelled EKE (Fig. 5f) which is highest close to 79° N and the daily velocity field (Supplementary Material: Movie S2) which shows much narrower velocity structures. Further, the section at 0° EW is less synoptic than the other sections presented in this study due to large time gaps between some stations (see caption of Fig. 5). Note that the watermass properties are not affected by the coarse temporal and spatial resolution.

Previous studies have reported eastward transport north of 79° 30'N at 0° EW (Marnela et al., 2013) variously related to the Molloy Hole (e.g. Hattermann et al., 2016). This is seen in the observations though not in the model average. The model study by Hattermann et al. (2016) described two branches of westward recirculation through Fram Strait, at 78.5° N and at 80° N. This agrees well with our synoptic section at 0° EW, both with the velocity field and, more conclusively, with the location of two salinity fronts (Fig. 5b/c). FESOM also shows two recirculation branches, which merge at 0° EW (Fig. 8c+d). Longterm averages of model output suggest that the mean current through 0° EW is southwestward (Hattermann et al., 2016; Kawasaki and Hasumi, 2016; Wekerle et al., 2017, and Fig. 5d in this study) and daily averages of the velocity field from FESOM show eddies advected southwestward (Supplementary Material: Movie S2).

### 3.3 The evolution of the EGC from northern Fram Strait to the Greenland Sea

In the synoptic section roughly perpendicular to the east Greenland shelfbreak at ~80.3° N (Section WT1), AW is only found in the central Fram Strait near 0° EW, some 130 km east of the Greenland shelfbreak (Fig. 3a). This is closer to the Svalbard shelfbreak than the Greenland shelfbreak. The deep $\theta$ maxima sampled west of 0° EW at WT1 have temperatures around

1°C, well below the temperature of AW, and salinities between 34.8 and 34.9 (Fig. 6a). This agrees with deep $\theta$ maxima from stations sampled between 82–83° N and 10–5° W in 2004 (Rudels et al., 2012), which, together with the transport measured there (Marnela et al., 2008) indicates that the AAW sampled at WT1 may be advected from the northwest along the east Greenland shelfbreak. Thus, the Arctic Ocean outflow of AAW sampled at 80.3° N is uninfluenced by directly recirculating AW west of 0° EW.

Salinity (Fig. 3b) increases strongly in the halocline over the upper 150 m. The density field (thin contour lines in Fig. 3b) closely follows the salinity field. At the mouth of Westwind Trough the temperature of the deep $\theta$ maximum is ~0.8°C.

Outside of the trough, two regions of southward flow were sampled (Fig. 3c). The local velocity maximum between 0–20 km offshore of the shelfbreak with relatively weak core velocities of -0.09 $\mathrm{m\,s^{-1}}$ is at a cross-shelfbreak distance where the shelfbreak EGC is found farther south. The broad southward flow between 5° W and 0° EW (30 km and 120 km), identified

as the Arctic Ocean outflow, is also visible in the modelled velocity field (Fig. 4b). Both bands of southward flow are highly barotropic and modelled EKE is negligible at WT1 (Fig. 4c).

In the Arctic Ocean outflow, at ~80.3° N (section WT1), the slope of the 27.8 $\mathrm{kg\,m^{-3}}$ isopycnal between 0° EW and the shelfbreak (Fig. 3b) is very weak (0.25 $\mathrm{m\,km^{-1}}$, corresponding to a baroclinic velocity of only 0.05 $\mathrm{m\,s^{-1}}$). In this respect the southward flow at WT1 is different from the well-defined baroclinic boundary current structure of the EGC farther south

commonly described in the literature. Likewise, the 2001–2009 FESOM model mean shows weak isopycnal slopes (Fig. 4). Thus we hypothesise that the southward flow at WT1 may not be a boundary current tied to the shelfbreak.

In the eight-year model average the AW reaches much closer to the shelfbreak at 80.3° N than in the synoptic section (Fig. 4a) and actually reaches the shelfbreak during 20% of the year, though it does not propagate into Westwind Trough. In the FESOM configuration used here (Wekerle et al., 2017), runoff is taken from the interannual dataset of Dai et al. (2009), which does not

take into account subglacial and submarine melting of the Greenland ice-sheet. This however may be crucial to represent the northeast Greenland shelf circulation correctly. A different freshwater input from Greenland would likely have effects both on the circulation in the troughs as well as watermass transport and transformation in the southward flow along the shelfbreak. It may thus impact the distance from the shelfbreak at which AW is found in the model. From comparison with the sparse observations available (this study, a synoptic section in Rudels et al. (2005) and the climatology in Schaffer et al. (2017))

we are inclined to trust the density and velocity field in FESOM in northern Fram Strait, but are more cautious about the distribution of AW. Thus, correctly modelled currents may advect the wrong water mass in the model, specifically AW may be simulated too far in the west.

Just 50 km farther to the south, at 79.6° N, AW is found merely 30 km offshore of the shelfbreak in a core between 150–450 m depth (Fig. 3a). The 27.8 $\mathrm{kg\,m^{-3}}$ isopycnal has a downward slope of 0.5 $\mathrm{m\,km^{-1}}$ toward the shelfbreak (this corresponds

to a baroclinic velocity of $0.1 \mathrm{~m\,s^{-1}}$), which has a greater similarity to the EGC structure farther south (Håvik et al., 2017) than the WT1 section. The offshore divergence of the isopycnals may be caused by AW intruding below, into and/or above the AAW layer at depth. The spreading apart of the isopycnals in the ambient AAW by intruding AW is likely a generic process (i.e. not just present in this synoptic section), taking place whenever AW meets AAW at depth with a distinct and strong

horizontal gradient in stratification. Intruding AW at depth has lower stratification consistent with the strong atmospheric cooling experienced relatively recently by the AW in the Nordic Seas boundary current loop. Some interleaving is present in the CTD profiles at the transition between AW and AAW 30 km from the shelfbreak (orange profile in Fig. 6b). Largely barotropic southward velocities (~$0.16 \mathrm{~m\,s^{-1}}$, Fig. 3c) are found just offshore of the shelfbreak.

While the isopycnal slope at 79.6° N in the synoptic section and the eight-year model average (Fig. 4) are similar to the

familiar boundary current structure of the EGC farther south, the core of the modelled southward velocities lies farther from the shelfbreak than in our synoptic section. The modelled daily average velocities (Supplementary Material: Movie S1) suggest that the main cause of high southward velocities near the shelfbreak are eddies passing through 79.6° N. The mean modelled EKE and velocity (Fig. 4b+c) show higher values at the same distance from the shelfbreak supporting this interpretation. This means that our observation may either have resolved the southward flowing rim of an eddy, or we sampled 79.6° N at a time

when the EGC was a boundary current and close to the shelfbreak. The latter is supported by the fact that in the model the southward flow at 79.6° N lies closer to the shelfbreak in summer than in winter (Fig. 8). Conversely, the upward sloping isopycnals seen below 200 m suggest the presence of an AW eddy in the synoptic section.

Another 80 km farther to the south, at 79° N, AW is found at ~200 m depth at the east Greenland shelfbreak though no AW is found on the east Greenland shelf (Fig. 3a). The $27.9 \mathrm{~kg\,m^{-3}}$ isopycnal undulates strongly, following the temperature

field. Whilst the isopycnals $< 27.8 \mathrm{~kg\,m^{-3}}$ are almost flat above 100 m depth in the deep Fram Strait (between 2.5° W and 2.5° E) they deepen toward the west. The downward sloping isopycnals (a slope of $0.75 \mathrm{~m\,km^{-1}}$ toward the shelfbreak for the $27.8 \mathrm{~kg\,m^{-3}}$ isopycnal corresponding to a baroclinic velocity of $0.15 \mathrm{~m\,s^{-1}}$) are located at a distance from the shelfbreak at which the shelfbreak EGC is found in mooring observations (e.g. Beszczynska-Möller et al., 2012) and our model average (Fig. 4b) and coincide with southward absolute geostrophic velocities (Fig. 3c). Thus this section shows the familiar structure

of the EGC as a baroclinic boundary current.

At 79° N there are two cores of southward velocities (Fig. 3c). We identify the core just offshore of the shelfbreak centred around 5° W (20 km) and reaching -$0.15 \mathrm{~m\,s^{-1}}$ as the shelfbreak EGC.

The modelled average temperature and velocity field are naturally smoother than the synoptic section but show the same general structure with AW subducting westward below PSW (Fig. 4a). The EKE at 79° N is much higher than at the sections sampled

to the north and south of this and has a peak where the EGC is found. This high variability can also be seen in the daily averages of the velocity field (Supplementary Material: Movie S1).

At the mouth of Norske Trough (76.6° N, i.e. another 270 km farther to the south along the shelfbreak), AW is found in a broad core between 100 and 350 m depth at and offshore of the shelfbreak (Fig. 3a). Inside of the trough a thin layer of AW is found between 200 and 250 m, i.e. above 320 m which is the depth of the shallowest sill between the shelfbreak and the

inner shelf near the NEGIS glaciers (Schaffer et al., 2017). The model also shows an AW layer within Norske Trough, both

in the eight-year average (Fig. 4a) and in the daily averages for 2009 (Supplementary Material: Movie S1). Thus, AW is able to propagate through Norske Trough to the termini of the NEGIS glaciers. However, the modelled AW layer is thicker inside Norske Trough than in the observations and thins eastward. Since this does also not agree with the temperature observations in Norske Trough reported in Schaffer et al. (2017), we again conclude that the model transports too much AW too far eastward.

The temperature of the synoptic deep $\theta$ maximum decreases from east to west and its depth increases (Fig. 6c). Observed salinities (Fig. 3b) are lowest at the surface and on the shelf. The density field largely follows the salinity field and isopycnals deepen toward the west (Fig. 3b). The 27.8 $\mathrm{kg\,m^{-3}}$ isopycnal has a downward slope of 1.66 $\mathrm{m\,km^{-1}}$ toward the west which corresponds to a baroclinic velocity of 0.33 $\mathrm{m\,s^{-1}}$.

Absolute geostrophic velocities on the shelf are northeastward whereas the shelfbreak EGC flows southwestward on the slope,
both in the observations and in the model, with high velocities (0.15–0.3 $\mathrm{m\,s^{-1}}$) throughout the water column (Fig. 3c). The core of the measured flow is located around 7° W (at 20 km) and reaches -0.26 $\mathrm{m\,s^{-1}}$. The EGC has a width of approximately 40 km and the observations show some surface intensification in its western half whereas the eastern part is more barotropic. The location and width of the shelfbreak EGC at NT1 agree well with Section 10 from Håvik et al. (2017), which is located ~30 km to the north of section NT1.

**4   Discussion**

In the following we will examine the evolution of the Arctic Ocean outflow to the EGC from north to south. The change in dynamics is addressed by examining the baroclinic and barotropic components of the southward flow. We will then discuss the transport along the shelfbreak, examining the different watermasses (e.g. DSOW transport) and compare this with observations of the EGC farther south. Finally we will draw inferences from our results about the circulation on the northeast Greenland
shelf.

**4.1   Formation of and transport in the EGC**

Both the observations and the model indicate that the recirculating AW first gets close to the east Greenland shelfbreak between the mouth of Westwind Trough at 80.3° N and 79.6° N. From our observations (Fig. 3) and the modelled velocity field of the AW layer in Fram Strait (Fig. 8c+d) we argue that this is likely to take place closer to 79.6° N than to WT1. The three sections
crossing the EGC downstream of WT1 show different stages of watermass transformation in the deep temperature maximum (Fig. 6): from AAW and AW located horizontally next to another at 79.6° N to successively greater mixing between the two until the deep temperature maximum is warmer than 2°C (and thus falls into the AW definition) at all stations sampled in section NT1. Successively more AW gets entrained into the core of the EGC from north to south, with the contribution of the AW endmember increasing from only 19±8 % at WT1 to 80±3 % at NT1 (Fig. 7c). This can also be seen in the north to south
increase of temperature, salinity and oxygen concentration, and the decrease of the depth and density of the deep temperature maximum within the core of the southward flow (Fig. 7a/b). The EGC at 79° N stands out as having the highest spread of values for all examined properties; a result of the strong ongoing stirring between the AAW transported in the EGC and recirculating

AW from the east, mixing then takes some more time to homogenize the water properties.

This high variability in watermass properties at 79° N is not merely synoptic, the modelled eight-year average EKE at 79° N (Fig. 4c) is significantly higher than at the sections crossing the east Greenland shelfbreak to the north and south. Modelled EKE is negligible at WT1; to the south of WT1, values increase and higher EKE values are found closer to the shelfbreak.

South of 79° N the EKE decreases again though the EKE maximum at NT1 is found closer to the shelfbreak than at 79° N, consistent with a "funnelling" of the southward flow. The high EKE values at 79° N would argue that the bulk of the eddy field of the recirculation crosses Fram Strait there. This is also seen in the model realisation of section 0° EW where the highest westward velocities coincide with the highest EKE values at ~79° N (Fig. 5d/f).

The transport of the southward flow along the shelfbreak varies between -0.9±0.75 Sv at WT1 and -4.0±0.75 Sv at NT1

and generally increases downstream (Fig. 7d). The exception is 79° N where the shelfbreak EGC transports only 1.1±1.2 Sv which is over 1.5 Sv less than at 79.6° N. (Since the section at 79.6° N did not sample the western edge of the southward flow at 79.6° N, transport through that section presents a minimum estimate). The mean southward core velocity increases from -0.08 m s$^{-1}$ at WT1 to -0.26 m s$^{-1}$ at NT1, again with section 79° N an exception (Fig. 7d). We have higher absolute errors at sections where a narrow current is sampled with relatively few stations. The relative error is especially high at 79°N where the

transport is low due to the highly variable synoptic flow field with flow reversals on small spatial scales. This also means that a closer station spacing would not necessarily make the flow field more interpretable and may lead to an even lower transport estimate. The transport through 79° N is low when compared with previous estimates of southward transport through 79° N (e.g. Fahrbach et al., 2001; de Steur et al., 2009, 2014; Schlichtholz and Houssais, 1999). Between 79° N and 78°50' N the summer mean EGC transport increases by ~2 Sv (de Steur et al., 2014) implying that a recirculation of this magnitude joins the

EGC between these two sections (this transport estimate includes, but is not restricted to AW). In winter, the transport increases by an additional ~3 Sv between the two latitudes, likely due to an intensification of the Greenland Sea Gyre (de Steur et al., 2014). Even the summer increase of 2 Sv is higher than the increase in our synoptic summer transport between 79.6° N and section NT1 (Fig. 7d). A more inclusive definition of the EGC, calculating transport at 79° N between 0° EW and 6° W (the latitudes used in de Steur et al., 2009, 2014) results in net northward transport in the synoptic section. Of course, de Steur et al.

(2009, 2014) report multi-year monthly means whereas our study is synoptic. It also has to be kept in mind that the station spacing of the moorings is wider than our station spacing and thus interpolation between moorings may remove much of the small scale variability that reduces transport in the synoptic section at 79° N. The majority of the synoptic transport measured at 79°N is synoptic AW transport. The synoptic transport of AAW, PSW and DW at 79°N is lower than at 79.6°N. This signal may be due to temporal variability or due to a change in pathway of these watermasses. AAW and PSW are found on the shelf

at 79° N where a surface intensified jet, which appears to be similar to the PSW Jet of Håvik et al. (2017), has a southward transport of 1.1 Sv (Fig. 3c).

91±5 % of the AW and the AAW have a density > 27.8 kg m$^{-3}$ which is the density definition of Denmark Strait Overflow Water (DSOW) (Fig. 7d). Transport of DSOW increases from -0.8 Sv at WT1 (80.3° N) to -2.5 Sv at NT1 (76.5° N). This may be explained by the gradual formation of the EGC as a baroclinic boundary current and the recirculating AW joining the

current. We use a lower boundary of 800 m in our definition, because Harden et al. (2016) demonstrated that north of Denmark

Strait aspiration across the strait's sill takes place down to ~800 m. If the deep 0°C isotherm is used as a lower boundary (as done in Håvik et al., 2017) our transport estimates increase by ~-0.1 Sv at each section. Remarkably, our transport estimate for DSOW at NT1 of -2.5 Sv agrees well with the -2.8±0.7 Sv average of the DSOW transport from 10 synoptic sections between 78° N and 68° N reported in Håvik et al. (2017) as well as with the mooring based annual mean of -2.5±0.2 Sv of Harden et al. (2016) for the EGC south of 68°30' N. This argues that the net DSOW transport along the east Greenland shelfbreak does not vary greatly between NT1 and Denmark Strait.

Since our definition of the width of the shelfbreak EGC follows Håvik et al. (2017), a comparison of the transport estimates is possible. Håvik et al. (2017) noted an increase in the shelfbreak EGC transport from 77.5° N to 74° N. Possibly due to the separation of the EGC into multiple branches south of this latitude, the transport begins to decrease. The transport of the shelfbreak EGC of their Section 10 (for location see Fig. 1) agrees with our estimate for NT1 though our velocities are significantly lower (Fig. 7d). Velocities at their Section 9 were closer to our value for NT1 though transport and current width were higher. Velocities and current widths measured by Håvik et al. (2017) were generally higher than those recorded in the present study. This is consistent if one assumes that the increase in isopycnal slope seen between WT1 and NT1 (Sect. 3.3) continues farther to the south. Another explanation could be the denser station spacing in Håvik et al. (2017) (5–7 km versus 10–20 km in our study). With a denser station spacing it is more likely to sample the location in the EGC with the highest velocity, thus making it more likely to arrive at a higher core velocity. Nevertheless, our study is able to extend the work by Håvik et al. (2017) northward of 79° N.

The definition used for the current width, which is based on a decrease of the core speed to 20 % of its maximum value, gives a southward increase from 20 km at WT1 to 40 km at NT1. However, this definition, though useful when comparing our data to the results of Håvik et al. (2017) farther south, is not able to capture the actual width of the entire southward flow, which decreases from the broad Arctic Ocean outflow we see at 80.3° N to the baroclinic boundary current at 79° N. This narrowing of the Arctic Ocean outflow to the shelfbreak EGC at 79° N can also be seen in the FESOM velocity fields (Fig. 8a+b) and other model studies (Hattermann et al., 2016; Kawasaki and Hasumi, 2016). It could be argued that the broad Arctic Ocean outflow and northern EGC in modelled velocity fields are an artefact of the multi-year averages portraying the mean of a meandering current. However, comparison with our observations at WT1 and daily averages of model velocities (Supplementary Material: Movie S1) suggests that the southward flow north of 79° N is indeed broad and not confined to the shelfbreak.

To better understand the transition of the EGC from a broad barotropic flow to a narrow baroclinic boundary current we examine the different components of the flow. The baroclinic velocity of the southward flow from the simple two layer estimate (Sect. 2.2) presented above increases from north to south. With the exception of the flow at 79.6° N (where the divergence of isopycnals at depth results in a more complicated baroclinic velocity field than captured in our simple two layer approximation), these baroclinic velocities agree well with the baroclinic velocities calculated from hydrography. Overall, this rough approximation gives some indication that the importance of the baroclinic (i.e. density driven) velocity component relative to the barotropic velocity component close to the shelfbreak increases from north to south. While the baroclinic estimate and absolute velocity components in the southward current at the shelfbreak show a north to south gradient (the baroclinic velocity increases 6-fold from WT1 to NT1), the barotropic velocity component does not show a clear latitudinal trend.

Even though the synoptic study presented here gives an indication that the EGC as a baroclinic boundary current is first observed at 79° N, continuing southward from there, it is only partly able to resolve the transition from a barotropic Arctic outflow north of 80° N to the density driven, baroclinic boundary current EGC seen south of 79° N. However, no previous studies have, to our knowledge, explicitly addressed this question, either for the EGC or more generally for subpolar boundary

currents. Further investigations are needed to establish if the local southward current maximum we observed at the shelfbreak at WT1 is a perennial feature and if either this, the southwestward current associated with the Polar Front at 0° EW, or both are the northward continuation of the EGC seen at 79° N. Multi-year averages from FESOM suggest that continuing north from 79° N the density front associated with warm recirculating AW and a band of high southward velocities are located farther east of the east Greenland shelfbreak (extending east of 0° EW north of 80° N) in the deep Fram Strait (Wekerle et al., 2017, and

Fig. 8c+d).

## 4.2   Impact of the EGC on the northeast Greenland shelf

We saw above that AW is far from the shelfbreak at Westwind Trough and a broad barotropic flow of Arctic outflow water flows southward between the AW and the trough's mouth. By the time the southward flow reaches Norske Trough a narrow and baroclinic EGC has formed in which the AW is mixed in with the ambient AAW, allowing the water >2°C to reach the

Trough's mouth. Thus AW can enter Norske Trough but not Westwind Trough. The depths of the deep temperature maximum, of the 1°C isotherm inside Norske Trough and of the 0.5°C isotherm inside Westwind Trough observed in this study all agree with the respective depths derived from a compilation of all CTD casts between 1979 and April 2016 (Schaffer et al., 2017). The evolution of the EGC documented above can explain this distribution of Atlantic derived waters on the east Greenland shelf.

Knee Water (KW) is defined as the sharp inflection in $\theta$ space of water close the freezing point (Bourke et al., 1987) and is formed in the Arctic Ocean by ice-ocean-atmosphere interaction (e.g. Rudels et al., 2005). The distribution of KW is an important indication of the shelf circulation. Both at 79° N (not shown) and inside Westwind Trough (Fig. 6a), KW is markedly absent at stations close to and on the east Greenland shelf. At NT1 the situation is reversed: KW is only found inshore of the shelfbreak (Fig. 6c). Observations close to the cavity of 79N Glacier show that the KW signal found inside Norske Trough is

eroded by isopycnal mixing with glacially modified water originating from both subglacial discharge and submarine melting (Schaffer, 2017). This leads to the hypothesis that KW is brought to the glaciers via Norske Trough and waters without the KW signature are then exported from the shelf via Westwind Trough. Thus KW and its absence can be used as a tracer for the shelf circulation. The distribution of KW in the EGC and on the shelf presented here support an anticyclonic circulation in the Atlantic derived water layer along the trough axes on the northeast Greenland shelf.

Only two of our sections extend far enough onto the east Greenland shelf to show part of the shelf circulation. The surface intensified current seen on the shelf at 79° N, between 7.5 and 10° W (-90 and -20 km, Fig. 3c), only transports small quantities of AW and may thus correspond to the PSW Jet described by Håvik et al. (2017) and seen in the hydrographic data of Nilsson et al. (2008) as far north as 79° N. The distance of the PSW Jet to the shelfbreak of ~50 km in our section and the peak velocity of -0.24 m s$^{-1}$ agrees with the distance from the shelfbreak and peak velocity reported in Håvik et al. (2017) between 71

and 68° N. Our synoptic transport in the PSW Jet (-1.1 Sv) is only slightly larger than the synoptic transports in the PSW Jet farther south (-0.54±0.28 to -0.83±0.27 Sv, Håvik et al., 2017). Unfortunately the sparsity of our shelf data and the velocity uncertainties inside Westwind Trough do not allow for a robust attribution of the cross-sectional flow seen inside Westwind Trough at ~60 km (Fig. 3c) as the northern continuation of the PSW Jet seen at 79° N. The PSW Jet is an important pathway for freshwater transport in the EGC current system (Håvik et al., 2017) and may impact exchanges between the EGC and the marine terminating glaciers. The presence of a strong density front between the fresh, cold PSW transported in the PSW Jet and the denser, warmer AW and AAW transported at the shelfbreak would create a barrier between the warmer water and the glacier termini, thus preventing these waters from contributing to submarine melt. The freshwater transport on the Greenland shelf may also provide a feedback mechanism between the glaciers themselves and the ocean, as described in Murray et al. (2010). The authors pose the theory that a decline in the East Greenland Coastal Current, a feature on the southeast Greenland shelf similar to the PSW Jet, can be linked to the influx of warmer water to the glacier termini and rapid speed-up of marine terminating glaciers. The subsequent influx of meltwater caused a strengthening and cooling of the East Greenland Coastal Current and was followed by a synchronized glacier deceleration. These possible links and processes call for a future detailed study of the northeast Greenland Shelf circulation.

## 5 Summary and conclusions

The maps of Fram Strait shown in Fig. 9 summarise the view of the recirculation supported by this analysis. The WSC advects AW northward in eastern Fram Strait where it looses contact with the atmosphere. Recirculating AW subducts underneath PSW and sea-ice on its way westward. The ice edge (Fig. 9a) appears on average to follow the location of AW subduction and the onset of the recirculation. It thus appears as if the dynamics of the recirculation set the seasonally relatively stable ice edge location in Fram Strait. In our synoptic summer survey along 0° EW no AW was found at 80.8° N, suggesting this latitude as the northern extent of the westward recirculation of AW in Fram Strait at that time. No AW is found within Westwind Trough or within 130 km east of its mouth. AW was first observed near the shelfbreak at 79.6° N and was found inside of Norske Trough, at a depth that would allow it to propagate to the terminus of the 79N glacier. Moreover, the deep $\theta$ maximum was above 2°C at all stations sampled at 76.5° N (at the mouth of Norske Trough), indicating that, at the depth of the deep temperature maximum, AAW had been completely mixed in with the recirculating AW. Figure 9a shows that, allowing for the synoptic variations, which of course do not show up in the long-term average of model output, the modelled AW layer thickness and our summer synoptic observations agree reasonably well in the southeastern Fram Strait. The most striking difference and something that should be investigated in the future is that water warmer than 2°C spreads farther northwestward in the model than observed there.

It is evident from the synoptic CTD sections along 79° N in Marnela et al. (2013), in Langehaug and Falck (2012) and in the present study as well as in the moored measurements of von Appen et al. (2016) that the synoptic view of 79° N differs substantially from the longterm mooring (e.g. Beszczynska-Möller et al., 2012) and model averages (e.g. Wekerle et al., 2017) which make the velocity field appear rather smooth (Fig. 4b). In contrast, Fig. 3c shows a qualitative picture of instantaneous

eddy variability. It is important to consider the highly variable structure of the flow field in order to reconcile measurements that at first seem counter-intuitive, such as northward flow in areas where the EGC is expected, with the overall circulation in Fram Strait. The daily velocity averages (Supplementary Material: Movie S1) and long-term EKE averages from FESOM (Fig. 4c) show that the very dynamic velocity structure at 79° N is not an artefact of our measurement technique. It rather is representative of the synoptic eddy field, a view that is typically lost in depictions of long-term (or even monthly) averages. The same is true for the surface intensified current on the shelf (the PSW Jet) that is seen in the synoptic section. This, too, is not discernible in the multi-year average of the FESOM velocity field though it is sometimes present in the daily averages. We think that it is representative that in our synoptic sections the boundary currents (WSC and EGC) instantaneously appear weaker than the eddies present in Fram Strait. This is supported by long term velocity measurements showing synoptic velocities in Fram Strait, likely associated with eddies, that are significantly higher than the time averaged velocity in the WSC (von Appen et al., 2016). The synoptic view presented here is also important for understanding the manifold processes, such as salt and heat transport to central Fram Strait and nutrient exchange between the surface layer and deeper watermasses, that are mediated by small scale features such as eddies. The small scale and highly variable structure of the velocity field in Fram Strait makes it essential to conduct both hydrographic surveys and model runs at an appropriate resolution to prevent aliasing. At the same time, it needs to be considered that any particular water sample taken in Fram Strait derives from this eddy field and may either have originated from inside or outside of transient eddies.

The Arctic Ocean outflow region between the northeast Greenland shelf and 0° EW is evident as a broad barotropic flow both in our synoptic section at 80.3° N (Fig. 3c) and in the velocity field from FESOM (Fig. 4b). From examination of the modelled velocity output we hypothesise that this Arctic Ocean outflow is at least partly topographically steered (see Fig. 7 in Wekerle et al., 2017). We propose that the evolution of the barotropic Arctic Ocean outflow to the baroclinic EGC is driven by the recirculation of AW in Fram Strait. As the recirculating AW reaches ever closer to the east Greenland shelfbreak, the Arctic Ocean outflow is restricted to an increasingly narrow band along the shelfbreak. At the same time the density difference between recirculating AW and waters of Arctic origin drives a baroclinic current. In northern Fram Strait, where the maximum westward extent of AW is located in central Fram Strait close to 0° EW, a baroclinic current associated with the Polar Front and the ice edge was described to merge with the EGC farther south (Schlichtholz and Houssais, 1999). This current was described as part of the EGC by Paquette et al. (1985). Farther south, the baroclinic boundary current EGC was also associated with the Polar Front, there located at the east Greenland shelfbreak as recirculating AW has spread farther west in Fram Strait. Here we argue that the EGC, Arctic Ocean outflow and AW recirculation are not separate but that the latter two combine to form the EGC. In a more global perspective, there are other boundary currents which do not follow a shelfbreak in their upstream part; these have to join the shelfbreak somehow. For example, different idealized models (Lighthill, 1969; Endoh, 1973; Suginohara, 1980) showed that barotropic and baroclinic Rossby waves from the ocean interior can explain the formation of western boundary currents. Seemingly eddies may play the same role as Rossby waves. We further presume that eddies in Fram Strait transport warm water to the western boundary which increases the along boundary transport.

Aspects of the circulation that require further study are the northern extent of the recirculation, the spatial distribution of AW between the shelfbreak near Westwind Trough and 0° EW, the circulation structure in the central Fram Strait north of 79° N,

with the possible role of the Molloy Hole, and the shelf circulation. A better knowledge of these would allow further reaching questions to be studied, e.g. the nutrient and freshwater fluxes to and from the Arctic, the role of the Nordic Seas and Arctic Ocean in deep water formation or the effect of the shelf circulation on submarine melt of Greenland glaciers. Ultimately all of these components link in with the larger question of how the Arctic is influenced by the changing climate.

5   To answer the question whether the EGC exists in northern Fram Strait, we note that the baroclinic boundary current does not exist in northern Fram Strait. Here, we take southward flow in a baroclinic boundary current along the shelfbreak as a defining feature of the EGC. By this definition, based on our evidence, we conclude that the EGC does not exist north of 79° N. It rather appears that the southward transport in northern Fram Strait is the Arctic Ocean outflow.

*Data availability.* Data is available under the references: CTD casts (Kanzow et al., 2017a, b), LADCP (von Appen et al., 2017) and
10   VMADCP (Kanzow and Witte, 2016).

*Author contributions.* CW lead the analysis of the model output. MER lead the analysis of the data and interpretation of the data and model output, as well as the write up of the paper. All authors contributed to each of these points.

*Competing interests.* The authors declare that they have no conflict of interest.

*Acknowledgements.* We wish to thank the captain and crew of RV *Polarstern*. We would also like to thank Torsten Kanzow and Janin
15   Schaffer for their contributions to the comprehensive data set that this study is based on. Further we thank Janin Schaffer for the kind use of the map of Fram Strait in Fig. 1 and for useful discussions on glacier-trough exchange. Support for this study was provided by the Deutsche Forschungsgemeinschaft (DFG) through the grant OGreen79 as part of the Special Priority Program (SPP)-1889 "Regional Sea Level Change and Society" (SeaLevel) and the Helmholtz Infrastructure Initiative FRAM. Data for this study was collected under grant number AWI-PS100_01.

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

**Table 1.** Watermass definitions after Rudels et al. (2005). Boundaries of potential temperature $\theta$ in $^\circ$C and potential density $\sigma_\theta$ in $\mathrm{kg\,m^{-3}}$ are given. $\sigma_\theta$ is potential density referenced to the sea surface.

| Watermass | Acronym | Definition |
|---|---|---|
| Polar Surface Water | PSW | $\sigma_\theta \leq 27.70, \theta \leq 0$ |
| warm Polar Surface Water | PSWw | $\sigma_\theta \leq 27.70, \theta > 0$ |
| Atlantic Water | AW | $27.70 < \sigma_\theta \leq 27.97, \theta > 2$ <br> $\sigma_\theta < 27.70, S > 34.92$ |
| Arctic Atlantic Water | AAW | $27.70 < \sigma_\theta \leq 29.97, 0 < \theta \leq 2$ |
| Deep Water | DW | $\sigma_\theta > 27.97$ |
| Denmark Strait Overflow Water | DSOW | $\sigma_\theta > 27.8$, depth $< 800$ m |

**Table 2.** Watermass endmember definitions for mixing calculations. Watermass acronyms as in Table 1, $\theta$ is potential temperature.

| Watermass | Salinity | $\theta$ |
|---:|---|---|
| DW | 34.93 | -0.9 |
| AW | 35.1 | 4.1 |
| AAW | 34.8 | 0.8 |
| PSW | 34.17 | -1.8 |

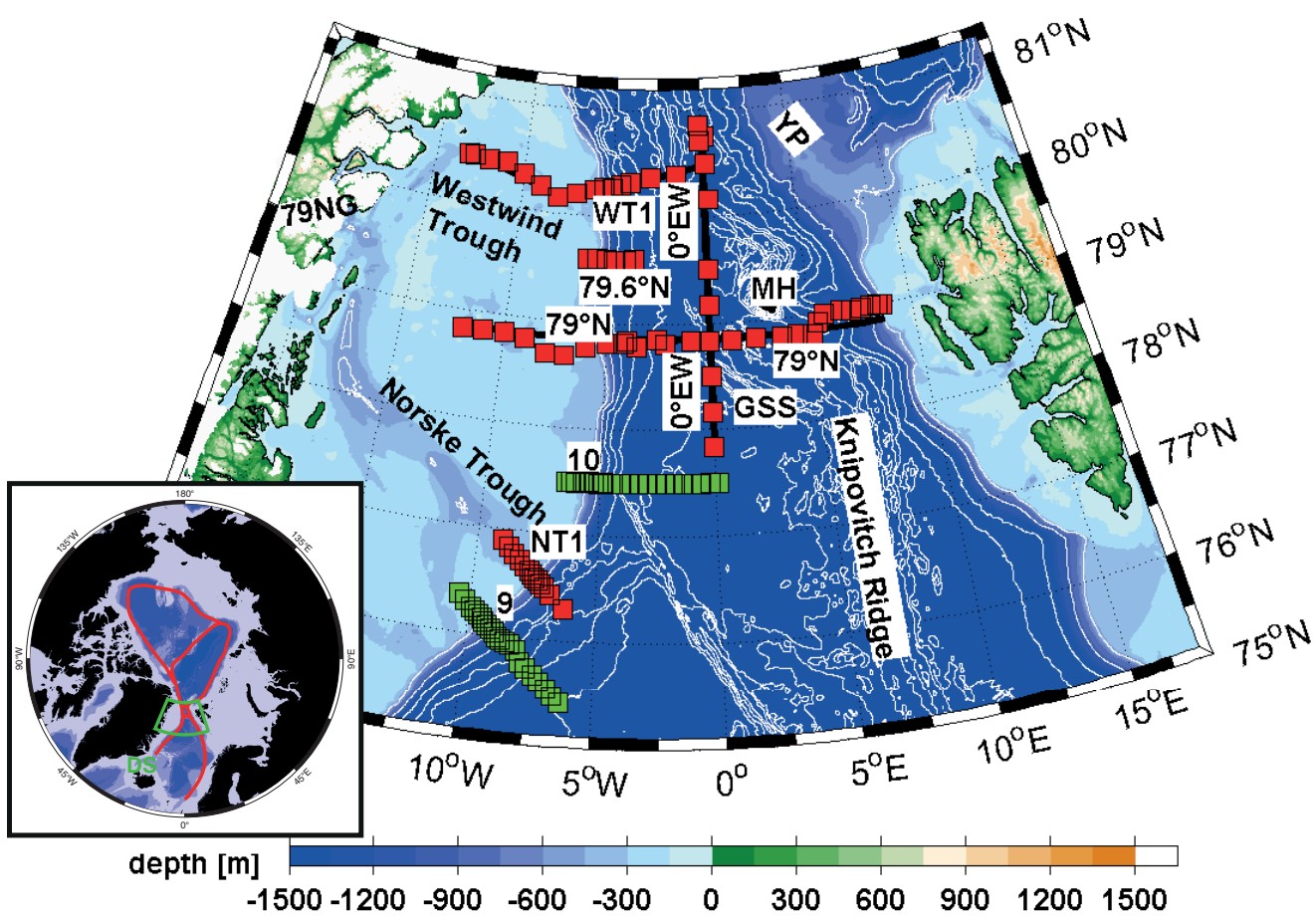

**Figure 1.** Map of Fram Strait between east Greenland and Svalbard. The inset shows AW and AAW pathways in the Nordic Seas and Arctic Ocean and the location of Denmark Strait (DS). Station locations are shown in red, the interpolated sections as bold black lines. The section names of this study are WT1, 79.6° N, ~79° N, NT1, 0° EW; also shown are Sections 9 and 10 of Håvik et al. (2017) in green. The locations of Norske and Westwind Trough, 79N Glacier (79N), Yermak Plateau (YP), Knipovitch Ridge, Greenland-Spitsbergen Sill (GSS) and Molloy Hole (MH) are also shown. Bathymetry from Schaffer et al. (2016).

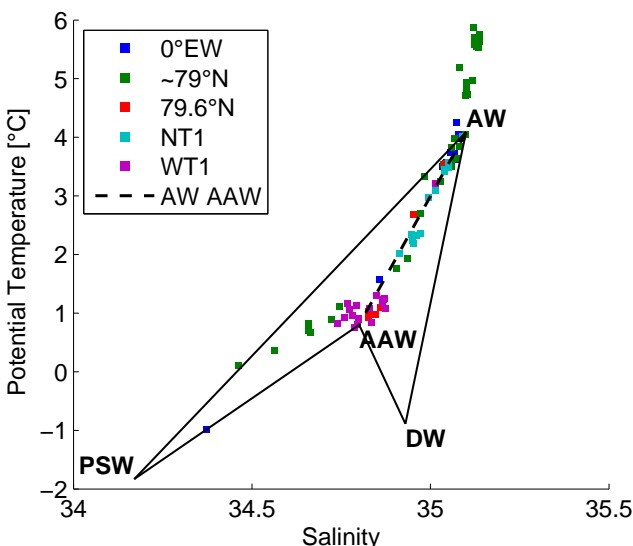

**Figure 2.** Mixing triangles of AW-AAW-PSW and AW-AAW-DW, abbreviations as in Table 1. Squares show the properties of the deep $\theta$ maximum at each station.

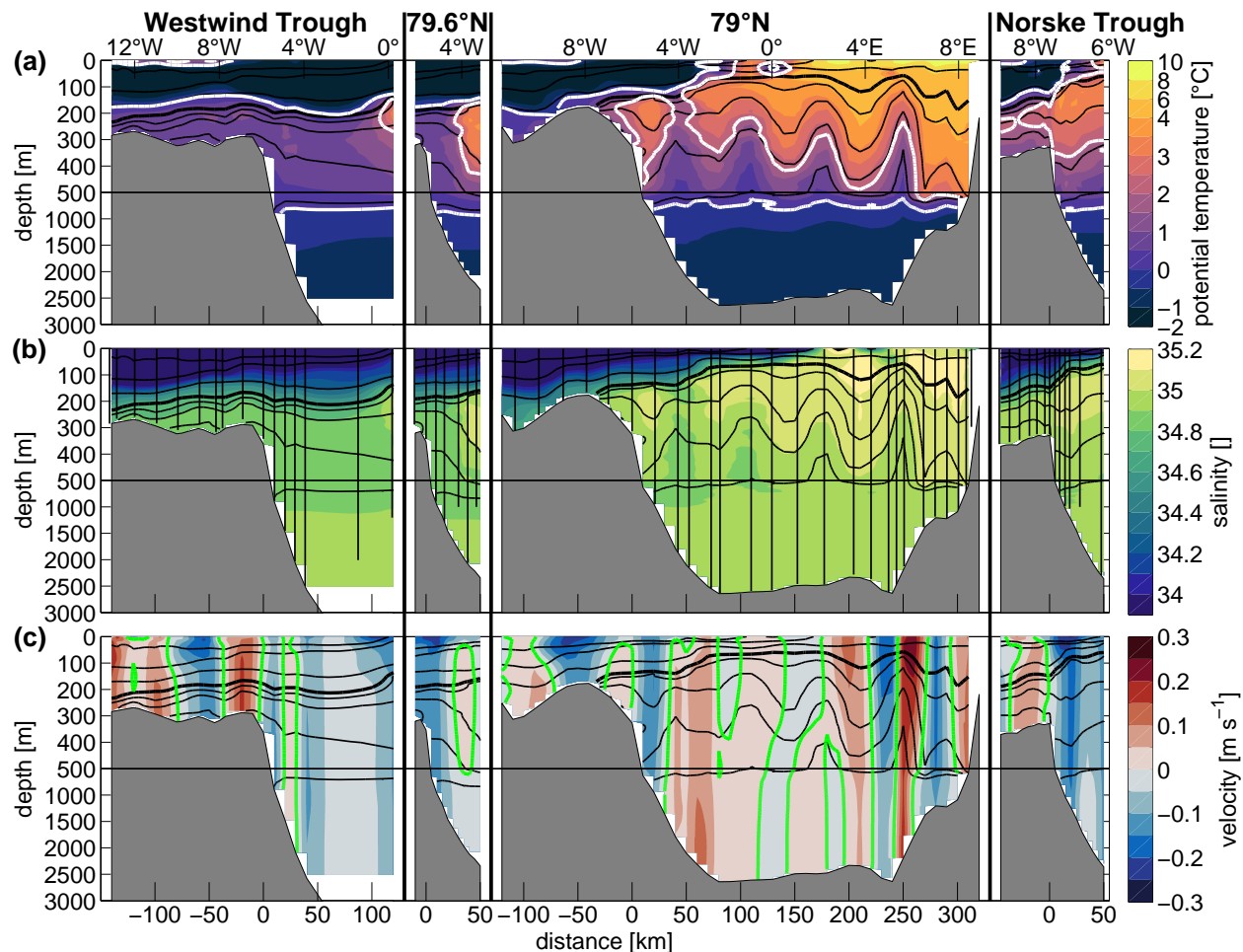

**Figure 3.** (a) Potential temperature, (b) salinity and (c) absolute geostrophic velocity for the sections crossing the east Greenland shelfbreak. Thin contours show potential density. The bold contour is the 27.8 kg m$^{-3}$ isopycnal; above the 26, 27 and 27.6 kg m$^{-3}$ isopycnals are shown, and below the density levels increase in 0.05 kg m$^{-3}$ steps up to 28 kg m$^{-3}$. Vertical black lines in (b) show station locations and depths. Please note the non-linear colourbars of salinity and temperature and that the y-axis changes scale at 500 m depth (black line). The white contours in (a) show the 2 ° C and 0 ° C isotherms. The green contours in (c) show the 0 m s$^{-1}$ isotach. Positive velocities are northward, negative velocities are southward. Section distance is 0 km at the east Greenland shelfbreak. At 79° N there is a gap of 11 days between the stations east and west of 2° E. Casts to the east were sampled within 6 days, casts to the west were sampled within 4 days.

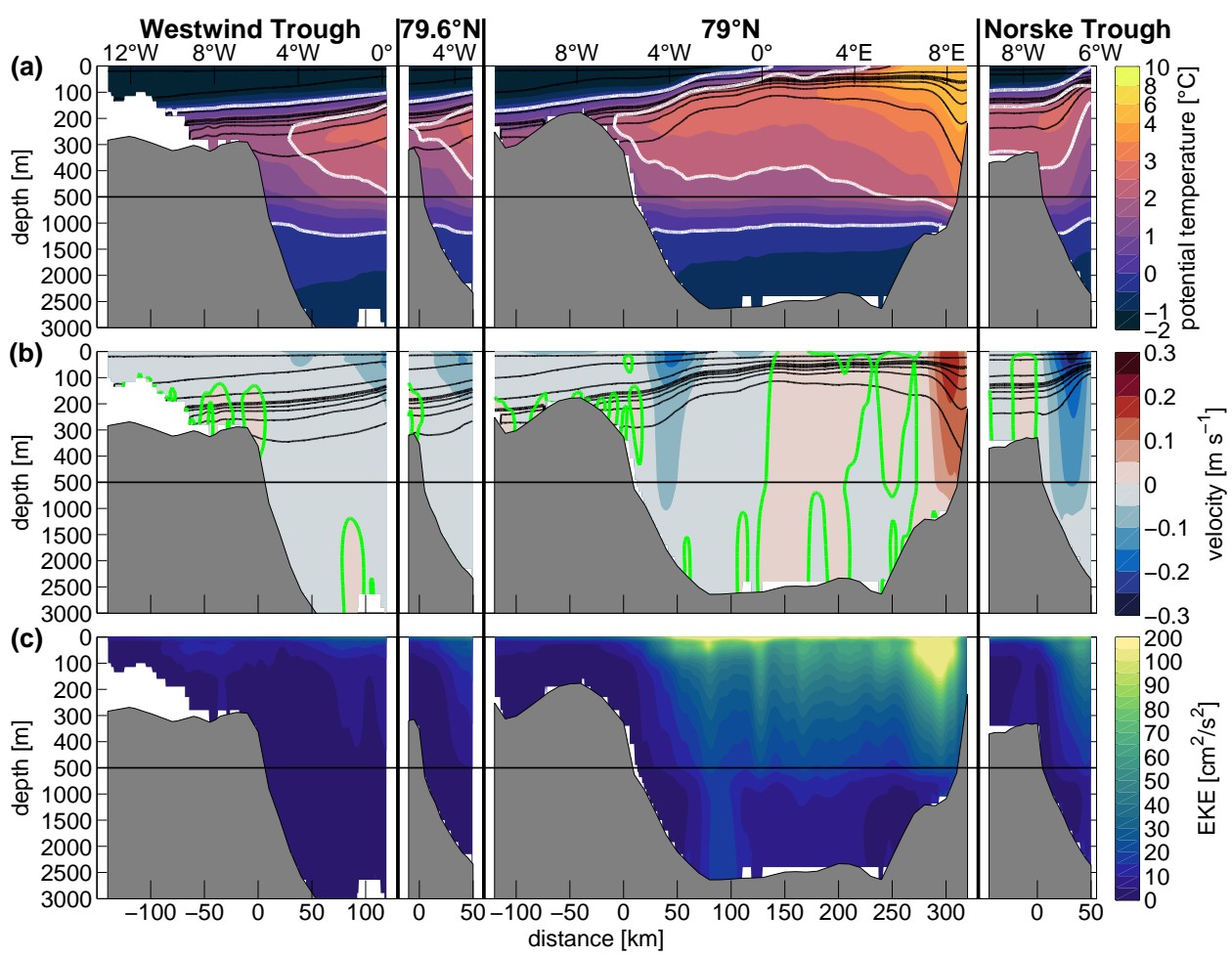

**Figure 4.** Eight year July/August/September average FESOM realizations of the sections crossing the east Greenland shelfbreak for (a) Potential temperature (as in Fig. 3a), (b) velocity (as in Fig. 3c) and (c) EKE.

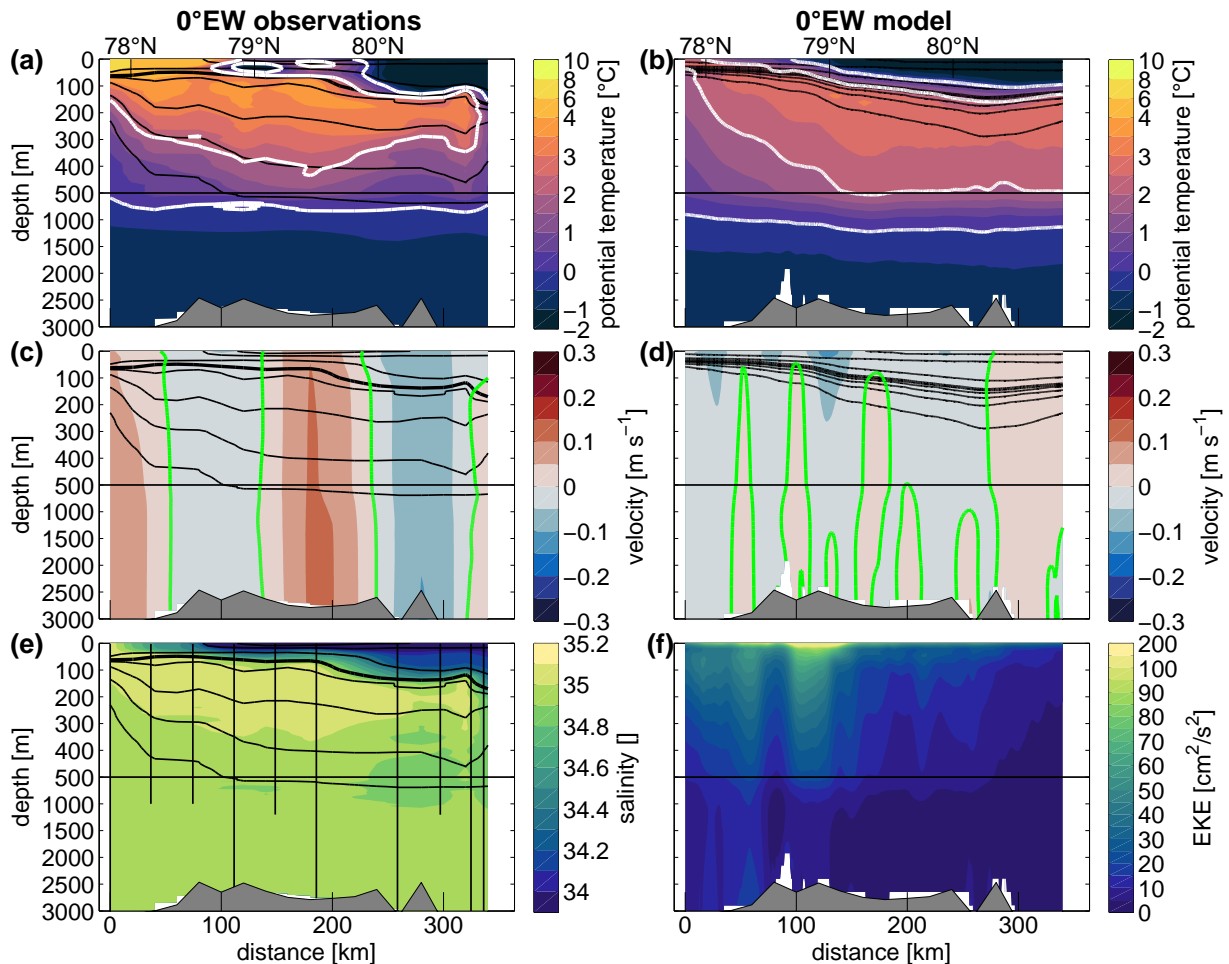

**Figure 5.** Left column: (a) Potential temperature, (c) absolute geostrophic velocity and (e) salinity as in Fig. 3 but for the section at $0°$ EW. The southernmost station was sampled last, 24 days after its northern neighbour. The next two stations were sampled 13 days after their northern neighbour. All remaining stations were occupied within 4 days. Right column: Eight year July/August/September average FESOM realizations of (b) Potential temperature, (d) velocity and (f) EKE as in Fig. 4 but for the section at $0°$ EW. Positive velocities are eastward, negative velocities are westward.

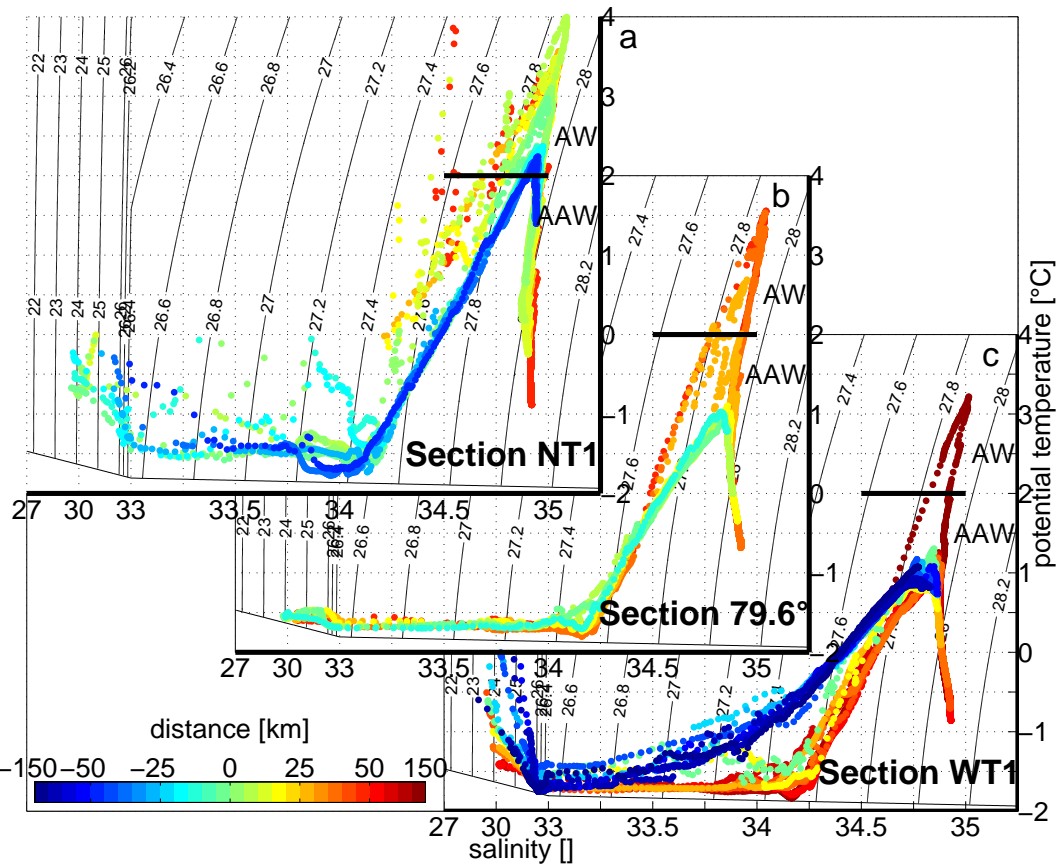

**Figure 6.** Potential temperature-salinity diagrams for three sections crossing the east Greenland shelfbreak (WT1, 79.6° N and NT1). Individual casts are colour coded depending on their distance to the east Greenland shelfbreak (positive = offshore). Please note that the x-axis changes scale at 33. The solid black line shows the watermass boundary between AW and AAW (see Table 1).

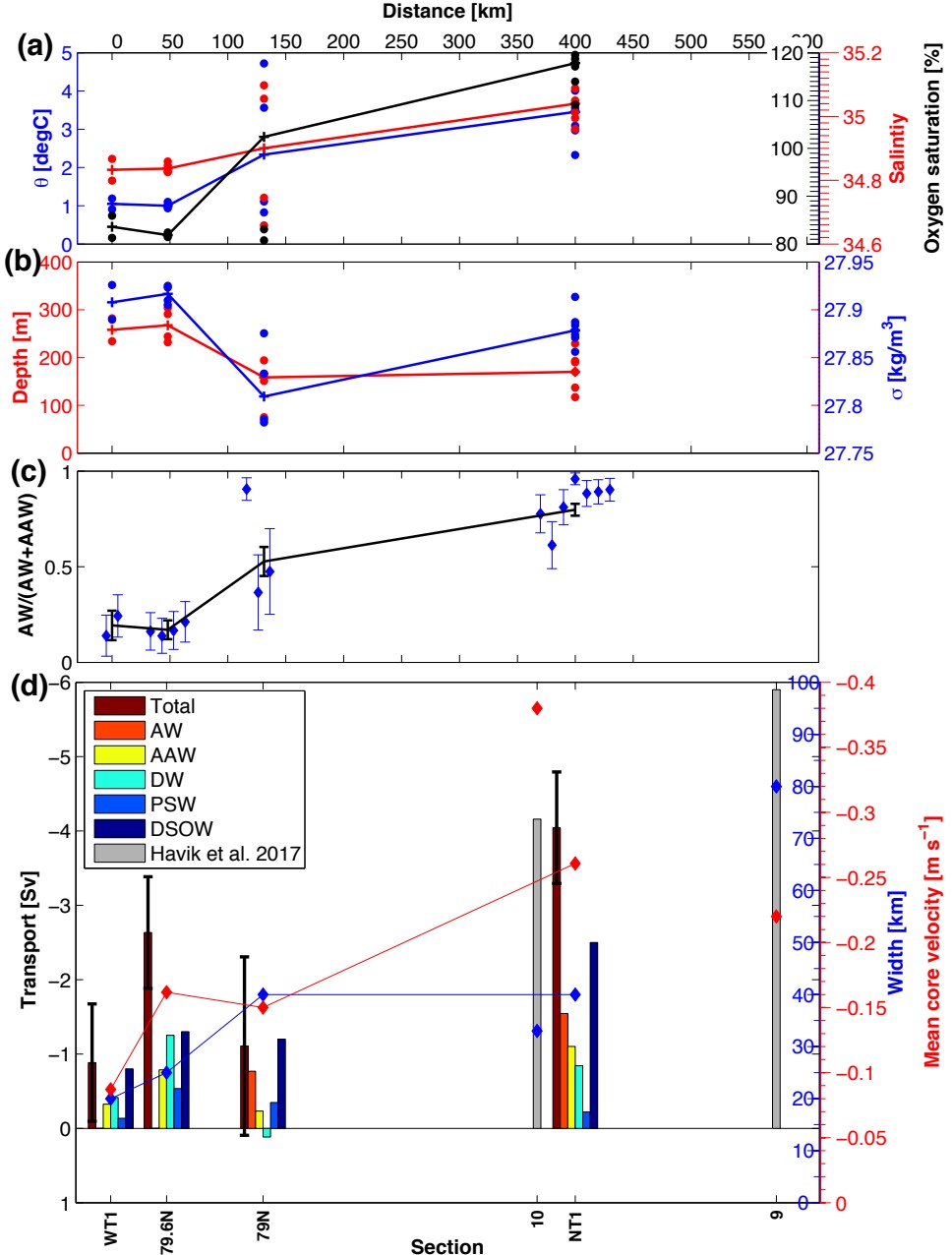

**Figure 7.** Properties of the θ maximum in the shelfbreak EGC from north to south: (a) potential temperature, salinity and oxygen saturation, (b) depth and potential density and (c) AW fraction as a function of AW+AAW for individual stations (blue, horizontally offset for clarity) and for the average at each section (black). Errorbars show the ±1 standard deviation. Transport, velocity and width of the shelfbreak EGC in Fram Strait as defined in Håvik et al. (2017) are shown in (d). Southward transport is negative. Watermass definitions as in Table 1. Downstream distance (in km) is 0 km at WT1 and follows the east Greenland shelfbreak southward. Values for Section 9 and 10 of Håvik et al. (2017) are taken from their paper.

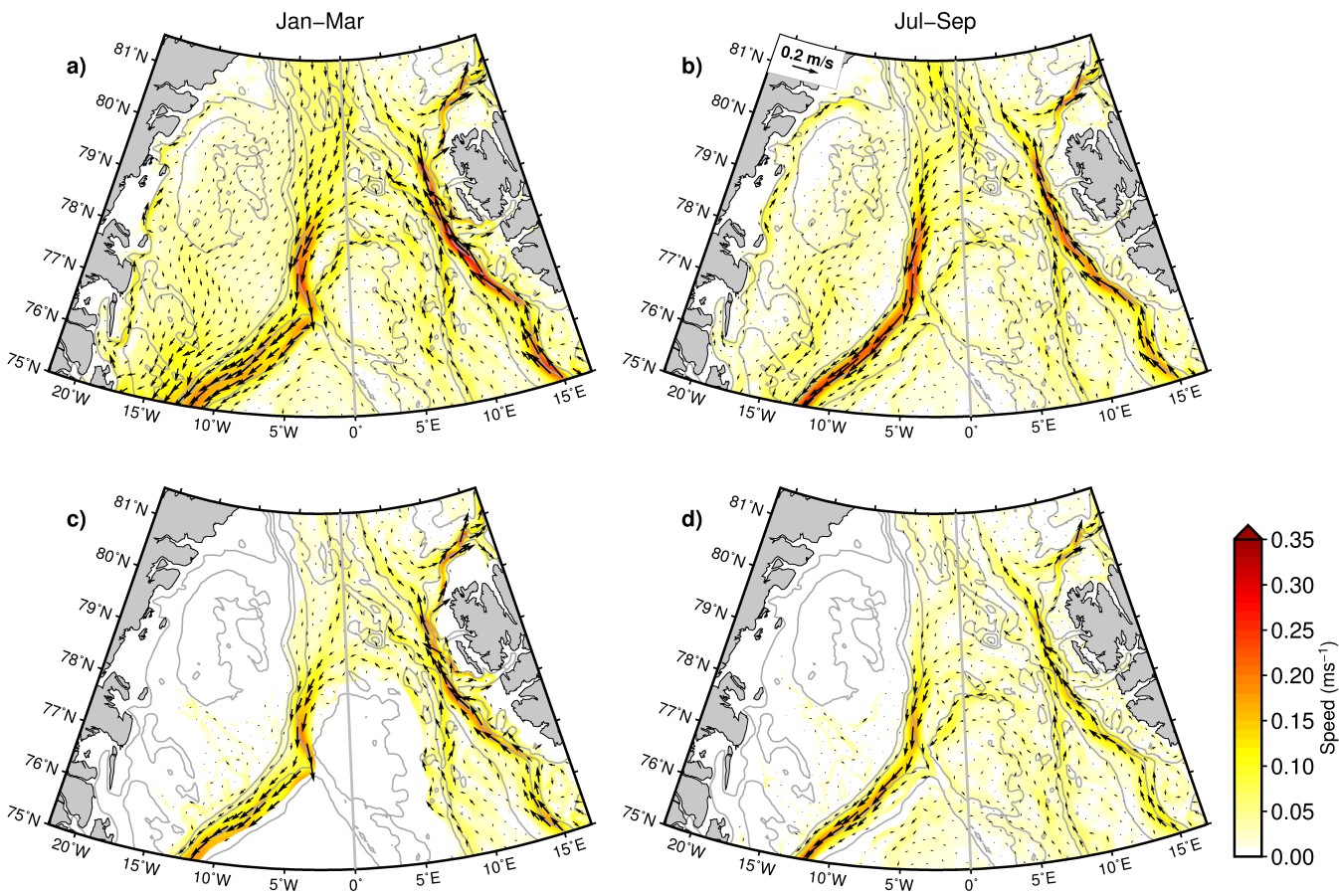

**Figure 8.** (Top) simulated velocity at 75 m depth and (Bottom) depth averaged simulated velocity in the AW layer (water warmer than 2°C) for the time periods (left) January-March and (right) July-September from FESOM. Thin gray lines are bathymetry, black arrows show current speed and direction and the coloured shading shows the speed.

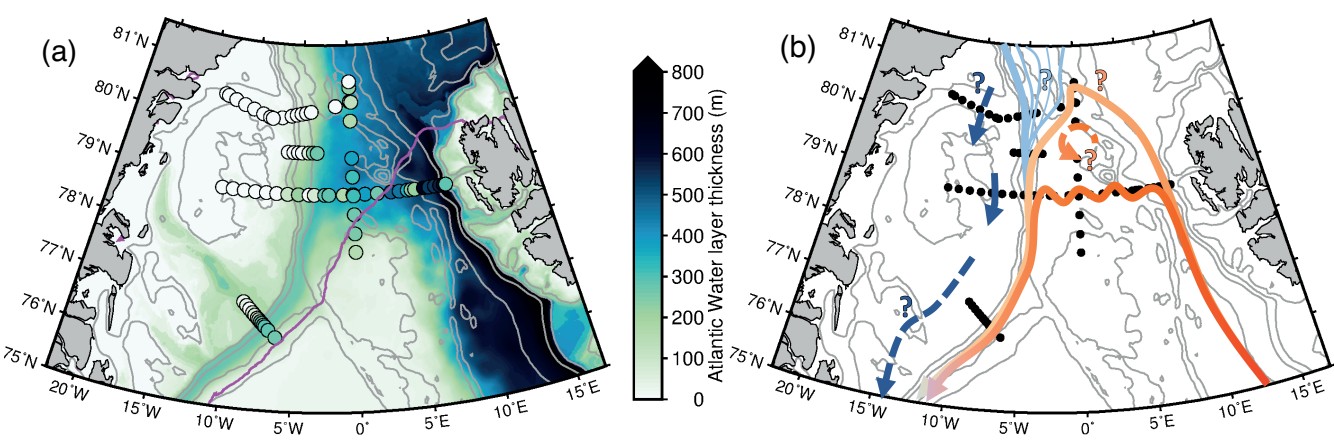

**Figure 9.** (a) AW-layer thickness in m. Shading shows the year-round 2000–2009 average from FESOM and coloured dots show the synoptic station data. Stations at which no AW was measured are shown in white. Thin gray lines are bathymetry, magenta is the modelled summer ice edge. (b) updated circulation scheme. Features we only speculate about are shown dashed and with a question mark.