# Peer review of "Does the East Greenland Current exist in northern Fram Strait?"

_Ocean Science, 2018_

## Referee Comment (RC1) · Anonymous Referee #1 · 28 Jun 2018

This is a really nice paper that describes an analysis of new field measurements combined with model output to examine the somewhat problematic question of the existence (or not) of the EGC north of Fram Strait. I only have a few minor comments for the authors.

p 1 L 11, suggest rephrase, e.g. the proportion of directly- or indirectly-sourced AW also increases fourffold, from ∼20% to ∼80%.

p 1 L 21, terminology: NEGIS vs. 79N glacier?

General question: I think the "further study" text (p 16 L 3) is rather narrowly expressed, meaning you confine your comments to your study region. There are surely larger questions concerning processes and continuity northwards of and outside the region?

[Figure]

I invite the authors to add something on this.

---

## Referee Comment (RC2) · Anonymous Referee #2 · 5 Jul 2018

os-2018-54

**Does the East Greenland Current exist in northern Fram Strait?**

by Maren Elisabeth Richter, Wilken-Jon von Appen, and Claudia Wekerle

In this manuscript a hydrographic/velocity survey and a high-resolution numerical model are used to investigate the circulation of Atlantic Water in Fram Strait, in particular to elucidate the recirculation from the West Spitsbergen Current to the East Greenland Current and to verify the existence of the East Greenland Current north of Fram Strait.

I think this manuscript is an important and valuable contribution to better understand the circulation in and north of Fram Strait. Despite continuous monitoring by moored instruments over several years, the circulation in this region is not well understood, in particular the western side of the strait. Lack of data on the Greenland shelf and north of the mooring transect have hampered previous investigations. The manuscript is for the most part well written and the figures are clear and easy to understand. I have a few comments that I hope the authors will take into consideration.

**General comments:**

I have two major concerns about the north-south section along the prime meridian (Fig. 5):
- the section is hardly synoptic
- the resolution is very coarse
The result is a very smooth section that is incapable of resolving eddies (e.g. page 15, line 6). Knowing the importance of the eddy field, this section is probably not a very good depiction of the circulation in Fram Strait. The section is useful as an indication of the northernmost presence of Atlantic Water, but the authors should take care not to make overinterpretations.

Another important source of error in the absolute geostrophic velocity calculation may stem from the tidal model. For example, a substantial component of the total error in the sections of Håvik *et al.* (2017) resulted from inaccurate bathymetry in the tidal model. Have you taken this into consideration?

**Specific comments:**

Page 1, line 11 and elsewhere:
For physical distance "farther" is more appropriate than "further".

Page 1, line 24:
The Norwegian Atlantic Current is also northward-flowing.

Page 2, line 17:
Some of the Atlantic Water transport into the Arctic Ocean likely flows through the Yermak Pass or around the Yermak Plateau as well (e.g. Koenig *et al.*, 2017).

Page 2, line 25:
Some of the upper part of the AW may also be transformed into a less saline surface layer instead of subducting beneath the PSW (e.g. Rudels *et al.*, 2015).

Page 2, line 32:
Please clarify which model suffers from a cold bias.

Page 3, line 14:

The reference to early observations of the EGC is a repeat from page 1, line 26.

Page 3, line 21:

Investigating the structure of the AW recirculation as a major objective of this study is not reflected in the title, only the second objective of investigating the EGC in northern Fram Strait.

Page 3, line 25 and elsewhere:

Is 0°EW a common way to refer to the prime meridian?

Page 3, line 32:

"Data" are usually considered plural.

Page 6, line 18:

Something is missing at the end of this sentence after "Svalbard".

Page 6, line 20:

It should be fields in plural.

Page 6, line 22:

A comma after unidirectional would make the sentence more clear.

Page 7, line 1:

The offshore branch of the WSC may be obscured by the presence of an eddy.

Page 7, line 12:

Southernmost should be written in one word.

Page 8, line 5:

It should be: "...southern limit of **the** Fram Strait recirculation..."

Page 9, line 31:

It should be: "...model average (Fig. 4) **are** similar to..."

Page 10, line 26:

The citation should not be in parentheses.

Page 10, line 27:

It is difficult to appreciate an increase in depth from Figure 6, in particular when isopycnals are missing from the θ-S plot.

Page 11, line 32:

It should be: **The** mean southward core velocity...

Page 12, line 4:

It should be: "...implying that **a** recirculation..."

Page 12, line 12:

It is not obvious why "missing" transport should be AAW, please clarify.

Page 12, line 15:

How large fractions of AW and AAW are denser than 27.8 kg/m$^3$?

Page 12, line 26:
It should be: "...noted an increase in **the** shelfbreak EGC transport..."

Page 13, line 9:
Is there a more effective way to demonstrate that the southward flow north of 79°N is indeed a broad flow rather than a meandering shelfbreak current than showing the mean velocity field?

Page 13, line 34:
Please repeat briefly which aspect of the EGC evolution can explain the distribution of Atlantic-origin water on the Greenland shelf.

Page 14, line 21:
Please elucidate how the PSW Jet can impact exchanges between the EGC and the marine terminating glaciers.

Page 14, line 23:
It appears that AW may be ventilated also within the EGC as it flows toward Denmark Strait (Våge *et al.*, 2018). As such, it is not correct to state that AW is last in contact with the atmosphere in Fram Strait.

Page 15, line 7:
It should be: "...structure of the flow field **it** is important..."

Page 15, line 13:
Can you use moored velocity records from Fram Strait (e.g. von Appen *et al.*, 2016) to support the conjecture that the boundary currents are weaker than the eddies?

Figure 6:
Please label the panels a-c and refer to specific panels when discussing the figure in the text. Contours of constant potential density would add information to the figure.

Figure 7:
Errorbars, in particular in transport, should be added to the figure.

Figure 9:
Instead of, or perhaps in addition to, Atlantic Water layer thickness, I think it would be effective to show AW and AAW fractions. This would clearly show how the composition of the EGC and the water masses in Fram Strait evolve along the circulation pathways.

**References**

Håvik L, Pickart RS, Våge K, Thurnherr AM, Beszczynska-Möller A, Walczowski W, von Appen WJ. 2017. Evolution of the East Greenland Current from Fram Strait to Denmark Strait: Synoptic measurements from summer 2012. *Journal of Geophysical Research: Oceans* : doi:10.1002/2016JC012 228.

Koenig Z, Provost C, Sennéchael N, Garric G, Gascard JC. 2017. The Yermak Pass Branch: A major pathway for the Atlantic Water north of Svalbard. *Journal of Geophysical Research: Oceans* **122**: 9332–9349, doi:10.1002/2017JC013 271.

Rudels B, Korhonen M, Schauer U, Pisarev S, Rabe B, Wisotzki A. 2015. Circulation and transformation of Atlantic water in the Eurasian Basin and the contribution of the Fram Strait inflow branch to the Arctic Ocean heat budget. *Progress in Oceanography* **132**: 128–152, doi:10.1016/j.pocean.2014.04.003.

Våge K, Papritz L, Håvik L, Spall MA, Moore G. 2018. Ocean convection linked to the recent ice edge retreat along east Greenland. *Nature Communications* **9**: doi:10.1038/s41 467–018–03 468–6.

von Appen WJ, Schauer U, Hattermann T, Beszczynska-Möller A. 2016. Seasonal cycle of mesoscale instability of the West Spitsbergen Current. *Journal of Physical Oceanography* **46**: 1231–1254, doi:10.1175/JPO–D–15–0184.1.

---

## Author Comment (AC1) · 2 Aug 2018

We have addressed the points raised by the reviewers and made changes to the manuscript accordingly. Please find our replies (in blue) to the reviewer comments (in black) in the document RepliesReferee1EGCMRichterWJvAppenCWekerle.pdf Page and line numbers refer to the marked up version of the manuscript found in EGCMRichterWJvAppenCWekerleDiff.pdf Both documents are in the supplement folder.

Please also note the supplement to this comment:
https://www.ocean-sci-discuss.net/os-2018-54/os-2018-54-AC1-supplement.zip

---

## Author Comment (AC2) · 2 Aug 2018

We have addressed the points raised by the reviewers and made changes to the manuscript accordingly. Please find our replies (in blue) to the reviewer comments (in black) in the document RepliesReferee2EGCMRichterWJvAppenCWekerle.pdf The page and line numbers refer to the marked up version of the manuscript EGCM-RichterWJvAppenCWekerleDiff.pdf Both documents can be found in the supplement folder.

Please also note the supplement to this comment:
https://www.ocean-sci-discuss.net/os-2018-54/os-2018-54-AC2-supplement.zip

---

## Author Response (AR1)

Dear Mr Hecht,

We have addressed the points raised by the reviewers and made changes to the manuscript accordingly. Please find our replies to the reviewer comments below in blue. Page and line numbers refer to the marked-up manuscript below. We hope that you and the reviewers are satisfied with the changes we made and can accept the manuscript for publication in OS.

Best Regards,

Maren Richter

**Anonymous Referee #1**

This is a really nice paper that describes an analysis of new field measurements combined with model output to examine the somewhat problematic question of the existence (or not) of the EGC north of Fram Strait. I only have a few minor comments for the authors.
We thank the reviewer for their very supportive review. We have addressed their comments as detailed below and hope that the manuscript can now be accepted.

p 1 L 11, suggest rephrase, e.g. the proportion of directly- or indirectly-sourced AW also increases fourfold, from ~20% to ~80%.
We thank the reviewer for helping to improve the abstract. We have rephrased the abstract accordingly. Page 1, line 11.

p 1 L 21, terminology: NEGIS vs. 79N glacier?
The NEGIS (North East Greenland Ice Stream) is a velocity structure in the Greenland ice shield, the 79N Glacier is one of the glaciers which discharges ice from this ice stream into the ocean. We thank the reviewers for pointing out that the term 79N Glacier is not mentioned in the introduction. We have amended this on page 1, line 22 and page 14, line 26 and believe that the terminology is now made sufficiently clear.

General question: I think the "further study" text (p 16 L 3) is rather narrowly expressed, meaning you confine your comments to your study region. There are surely larger questions concerning processes and continuity northwards of and outside the region? I invite the authors to add something on this.

We thank the reviewer for this suggestion and have broadened the "further study" paragraph on page 17, line 3 to 6. We hope that the paragraph now reflects the many connections our study has to other fields of research whilst remaining concise.

Reviewer 2:

In this manuscript a hydrographic=velocity survey and a high-resolution numerical model are used to investigate the circulation of Atlantic Water in Fram Strait, in particular to elucidate the recirculation from the West Spitsbergen Current to the East Greenland Current and to verify the existence of the East Greenland Current north of Fram Strait.
I think this manuscript is an important and valuable contribution to better understand the circulation in and north of Fram Strait. Despite continuous monitoring by moored instruments over several years, the circulation in this region is not well understood, in particular the western side of the strait. Lack of data on the Greenland shelf and north of the mooring transect have hampered previous

investigations. The manuscript is for the most part well written and the figures are clear and easy to understand. I have a few comments that I hope the authors will take into consideration.

We thank the reviewer for their supportive review. Our replies to the detailed and constructive suggestions and comments are found below.

General comments:
I have two major concerns about the north-south section along the prime meridian (Fig. 5):
- the section is hardly synoptic
- the resolution is very coarse
The result is a very smooth section that is incapable of resolving eddies (e.g. page 15, line 6). Knowing the importance of the eddy field, this section is probably not a very good depiction of the circulation in Fram Strait. The section is useful as an indication of the northernmost presence of Atlantic Water, but the authors should take care not to make overinterpretations.

We are confused about the reference to page 15 line 6 as this does not deal with the section along 0\degree EW. In our manuscript we point out both concerns the reviewer has with the section (page 8 line 19 and 22. We believe that it is nonetheless interesting to show this section and that the reader can form their own opinion on how far they whish to follow our suggestions as to the interpretation of the section. As the reviewer points out the section is still an important indication on how far north recirculating AW can reach at 0°EW. We have softened some of our conclusions regarding the section (page 7, line 25-26, page 8, line 16, 18, 22-24, 28), hoping that the section now no longer causes the reviewer concern.

Another important source of error in the absolute geostrophic velocity calculation may stem from the tidal model. For example, a substantial component of the total error in the sections of H°avik *et al.* (2017) resulted from inaccurate bathymetry in the tidal model. Have you taken this into consideration?

Thank you for your comment. We have added a formal error estimation for the transport calculation following Havik et al. (2017) and Sutherland (2009) (page 5, line 11-15 and page 12, line 10 and 16 to 18) including information on how we dealt with the tidal model.

Specific comments:
Page 1, line 11 and elsewhere:
For physical distance "farther" is more appropriate than "further".
Thank you for pointing this out, we have changed this throughout.

Page 1, line 24:
The Norwegian Atlantic Current is also northward-flowing.
We have rephrased the sentence and hope that it is now made clear that both the WSC and the Norwegian Atlantic current are northward flowing (page 1, line 26 and page 2, line 1).

Page 2, line 17:
Some of the Atlantic Water transport into the Arctic Ocean likely flows through the Yermak Pass or around the Yermak Plateau as well (e.g. Koenig *et al.*, 2017).
Thank you for making us aware of this paper, we have added a note on the flow through Yermak Pass (page 2, line 20-22).

Page 2, line 25:
Some of the upper part of the AW may also be transformed into a less saline surface layer instead of subducting beneath the PSW (e.g. Rudels *et al.*, 2015).
We are aware of the publication by Rudels et al 2015 and other authors studying the Arctic Ocean halocline. We none the less feel that to explicitly discuss the AW transformation in the AO further

than the sentence which we added on page 2, line 10, would disrupt the flow of the introduction and lead too far away from the focus of our study.

Page 2, line 32:
Please clarify which model suffers from a cold bias.
We have added the information as suggested (page 3, line 2).

Page 3, line 14:
The reference to early observations of the EGC is a repeat from page 1, line 26.
Removed (page 3, line 19).

Page 3, line 21:
Investigating the structure of the AW recirculation as a major objective of this study is not reflected in the title, only the second objective of investigating the EGC in northern Fram Strait.
We have rephrased the statement (page 3, line 26-27).

Page 3, line 25 and elsewhere:
Is 0◦EW a common way to refer to the prime meridian?
We have seen 0°EW used before but have now introduced the term on page 3, line 30 to avoid confusion.

Page 3, line 32:
"Data" are usually considered plural.
We are aware that Data is plural but since the term is widely used as though it were singular we believe that it is acceptable to keep it like this as long as it is consistent throughout the manuscript.

Page 6, line 18:
Something is missing at the end of this sentence after "Svalbard".
Thank you for pointing this out, we have inserted the missing word on page 6, line 27.

Page 6, line 20:
It should be fields in plural.
Changed, page 6, line 28.

Page 6, line 22:
A comma after unidirectional would make the sentence more clear.
Thank you, we have followed your suggestion, page 6, line 30.

Page 7, line 1:
The offshore branch of the WSC may be obscured by the presence of an eddy.
We have added a sentence to that effect on page 7, line 12.

Page 7, line 12:
Southernmost should be written in one word.
We have corrected this throughout the document.

Page 8, line 5:
It should be: "...southern limit of the Fram Strait recirculation..."
Changed on page 8, line 14.

Page 9, line 31:
It should be: "...model average (Fig. 4) are similar to..."
Changed on page 10, line 9.

Page 10, line 26:
The citation should not be in parentheses.
Changed on page 11, line 4.

Page 10, line 27:
It is difficult to appreciate an increase in depth from Figure 6, in particular when isopycnals are missing from the T-S plot.
We agree that a change in depth is not directly visible in Figure 6 and have added isopycnals to the figure as suggested. Isopycnals are also shown in Figure 3, where a change in depth is more easily visible.

Page 11, line 32:
It should be: The mean southward core velocity...
Changed on page 12, line 13.

Page 12, line 4:
It should be: "...implying that a recirculation..."
Changed on page 12, line 20.

Page 12, line 12:
It is not obvious why "missing" transport should be AAW, please clarify.
We have rephrased the paragraph on page 12, line 29-32 and hope that our argument is clearer now.

Page 12, line 15:
How large fractions of AW and AAW are denser than 27.8 kg/m$_3$?
We have calculated the fraction and have included it in the manuscript on page 12, line 34.

Page 12, line 26:
It should be: "...noted an increase in the shelfbreak EGC transport..."
Changed on page 13, line 10.

Page 13, line 9:
Is there a more effective way to demonstrate that the southward flow north of 79◦N is indeed a broad flow rather than a meandering shelfbreak current than showing the mean velocity field?
We are unsure what the reviewer means. The Figure referred to here is in the supplementary materials and is a movie in gif format which shows the variability at WT1 over an entire year. We now refer to S1 and S2 as movies instead of as figures to make this clear to the reader.

Page 13, line 34:
Please repeat briefly which aspect of the EGC evolution can explain the distribution of Atlantic-origin water on the Greenland shelf.
We have added a paragraph on page 14, line 14-17 repeating the main points from the section above and hope that this makes the following section clearer to the reader.

Page 14, line 21:
Please elucidate how the PSW Jet can impact exchanges between the EGC and the marine terminating glaciers.
Thank you for pointing out that we need more information on this. We have added a paragraph on page 15, line 8-16 on the shelf processes connected with the PSW Jet, EGC and the glacier termini.

Page 14, line 23:

It appears that AW may be ventilated also within the EGC as it flows toward Denmark Strait (V°age *et al.*, 2018). As such, it is not correct to state that AW is last in contact with the atmosphere in Fram Strait.

We have changed our statement on page 15, line 19 accordingly but feel that including information as in Vage et al 2018 would blur the focus of our argument and would not add to the text flow. The paper by Vage describes evidence for a process limited to a specific region far south of our study area.

Page 15, line 7:

It should be: "...structure of the flow field it is important..."

We think that perhaps the sentence structure was not quite clear, thus leading to this misunderstanding. We have rephrased the sentence on page 16, line 3 and hope that it is now easier to understand.

Page 15, line 13:

Can you use moored velocity records from Fram Strait (e.g. von Appen *et al.*, 2016) to support the conjecture that the boundary currents are weaker than the eddies?

Thank you for this suggestion, we have included this in our argument on page 16, line 11-13.

Figure 6:

Please label the panels a-c and refer to specific panels when discussing the figure in the text. Contours of constant potential density would add information to the figure.

We have added the labels a-c, references to these in the text, and isopycnals in the figure as suggested.

Figure 7:

Errorbars, in particular in transport, should be added to the figure.

We have added errorbars to the transport estimates as suggested.

Figure 9:

Instead of, or perhaps in addition to, Atlantic Water layer thickness, I think it would be effective to show AW and AAW fractions. This would clearly show how the composition of the EGC and the water masses in Fram Strait evolve along the circulation pathways.

We purposefully do not show AW and AAW fractions from the Model since the FESOM model has a salinity bias and the watermass definitions include a salinity/density criterion. We cannot make sure that "AW" or "AAW" in the Model and our measurements show the same thing. Using a temperature threshold only would mean that we can no longer distinguish AAW and PSW. Thus, it does not make sense to compare the measured fractions with the model.

[revised manuscript text omitted]